# An image reconstruction framework for characterizing initial visual encoding

**Ling-Qi Zhang\*, Nicolas P Cottaris, David H Brainard**

Department of Psychology, University of Pennsylvania, Philadelphia, United States

**Abstract** We developed an image-computable observer model of the initial visual encoding that operates on natural image input, based on the framework of Bayesian image reconstruction from the excitations of the retinal cone mosaic. Our model extends previous work on ideal observer analysis and evaluation of performance beyond psychophysical discrimination, takes into account the statistical regularities of the visual environment, and provides a unifying framework for answering a wide range of questions regarding the visual front end. Using the error in the reconstructions as a metric, we analyzed variations of the number of different photoreceptor types on human retina as an optimal design problem. In addition, the reconstructions allow both visualization and quantification of information loss due to physiological optics and cone mosaic sampling, and how these vary with eccentricity. Furthermore, in simulations of color deficiencies and interferometric experiments, we found that the reconstructed images provide a reasonable proxy for modeling subjects' percepts. Lastly, we used the reconstruction-based observer for the analysis of psychophysical threshold, and found notable interactions between spatial frequency and chromatic direction in the resulting spatial contrast sensitivity function. Our method is widely applicable to experiments and applications in which the initial visual encoding plays an important role.

## Editor's evaluation

This rigorous computational study simulates the sampling of the visual image by cone photoreceptors in the human eye, and explains how the image content can be reconstructed from those cone signals. The authors show that a number of properties of the human retina and of human perception are predicted from these simulations. Their modeling framework also serves to unify previous treatments and invites extension to subsequent stages of visual processing.

\*For correspondence:
lingqiz@sas.upenn.edu

## Introduction

Visual perception begins at the retina, which takes sensory measurements of the light incident at the eyes. This initial representation is then transformed by computations that support perceptual inferences about the external world. Even these earliest sensory measurements, however, do not preserve all of the information available in the light signal. Factors such as optical aberrations, spatial and spectral sampling by the cone mosaic, and noise in the cone excitations all limit the information available downstream.

One approach to understanding the implications of such information loss is ideal observer analysis, which evaluates the optimal performance on psychophysical discrimination tasks. This allows for quantification of the limits imposed by features of the initial visual encoding, as well as predictions of the effect of variation in these features (*Geisler, 1989*; *Geisler, 2011*). Ideal observer analysis separates effects due to the visual representation from inefficiencies in the processes that mediate the discrimination decisions themselves. Such analyses have often been applied to analyze performance for simple artificial stimuli, assuming that the stimuli to be discriminated are known exactly (*Banks et al., 1987*; *Davila and Geisler, 1991*) or known statistically with some uncertainty (*Pelli, 1985*;

*Geisler, 2018*). The ideal observer approach has been extended to consider decision processes that learn aspects of the stimuli being discriminated, rather than being provided with these a priori, and extended to handle discrimination and estimation tasks with naturalistic stimuli (*Burge and Geisler, 2011*; *Burge and Geisler, 2014*; *Singh et al., 2018*; *Chin and Burge, 2020*; *Kim and Burge, 2020*). For a recent review, see *Burge, 2020*; also see *Tjan and Legge, 1998* and *Cottaris et al., 2019*; *Cottaris et al., 2020*.

It is generally accepted that the visual system has internalized the statistical regularities of natural scenes, so as to take advantage of these regularities for making perceptual inferences (*Attneave, 1954*; *Field, 1987*; *Shepard, 1987*; *Knill et al., 1996*). This motivates interest in extending ideal observer analysis to apply to fully naturalistic input, while incorporating the statistical regularities of natural scenes (*Burge, 2020*). Here, we pursue an approach to this goal that, in addition, extends the evaluation of performance to a diverse set of objectives.

We developed a method that, under certain assumptions, optimally reconstructs images from noisy cone excitations, with the excitations generated from an accurate image-computable model of the front end of the visual system (*Cottaris et al., 2019*; *Cottaris et al., 2020*). (We use the term 'image-computable' here in contrast with observer models that operate on abstract and/or hypothetical internal representations.) The image reconstruction approach provides us with a unified framework for characterizing the information loss due to various factors in the initial encoding. In the next sections, we show analyses that: (1) use image reconstruction error as an information metric to understand the retinal mosaic 'design' problem, with one example examining the implications of different allocations of retinal cone types; (2) allow both visualization and quantification of information loss due to physiological optics and cone mosaic sampling and how this varies with eccentricity, as well as with different types of color deficiency; (3) combine the image reconstruction approach with analysis of

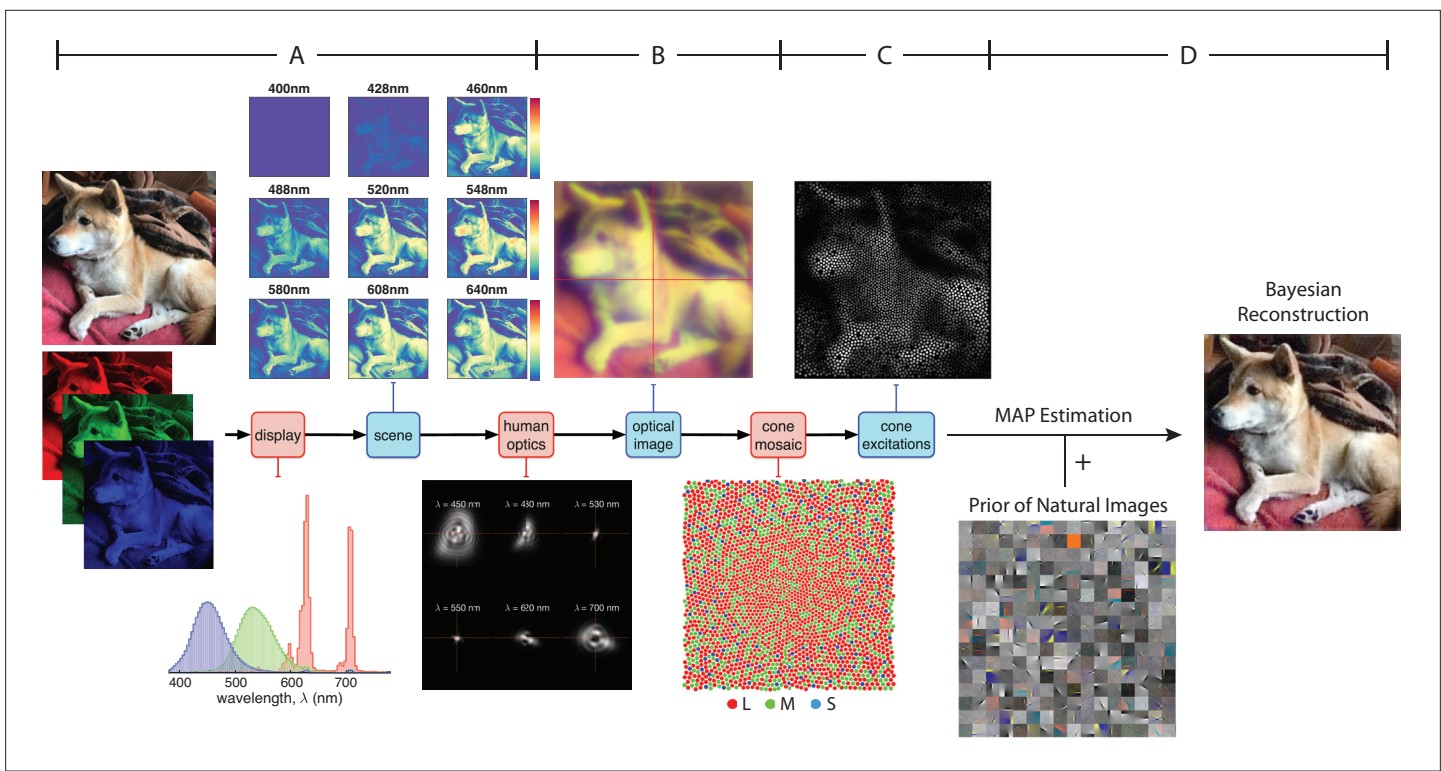

**Figure 1.** Model of the initial visual encoding and Bayesian reconstruction from cone mosaic excitation. (**A**) The visual stimulus, in our case a natural image in RGB format, is displayed on a simulated monitor, which generates a hyperspectral scene representation of that image. (**B**) The hyperspectral image is blurred with a set of wavelength-dependent point-spread functions typical of human optics. We also account for spectral transmission through the lens and the macular pigment. This process produces the retinal image at the photoreceptor plane. (**C**) The retinal image is then sampled by a realistic cone mosaic, which generates cone excitations (isomerizations) for each cone. The trial-by-trial variability in the cone excitations is modeled as a Poisson process. (**D**) Our Bayesian reconstruction method takes the pattern of cone excitations as input and estimates the original stimulus (RGB image) based on the likelihood function and a statistical model (prior distribution) of natural images (see Materials and methods).

psychophysical discrimination, thus providing a way to incorporate into such analyses the assumption that our visual system takes into account the statistical regularities of natural images.

## Results

We developed a Bayesian method to reconstruct images from sensory measurements, which we describe briefly here (see Materials and methods for details). We begin with a forward model that expresses the relation between an image and its visual representation at a well-defined stage in the visual pathway. Here that stage is the excitations of the photoreceptors of the retinal cone mosaic, so that our model accounts for blur in retinal image formation, spatial and spectral sampling by the cone mosaic, and the noise in the cone excitations. The approach is general, however, and may be applied to other sites in the visual pathways (see e.g. *Naselaris et al., 2009*; *Parthasarathy et al., 2017*). Our forward model is implemented within the open-source software package ISETBio (isetbio.org; *Figure 1A–C*) which encapsulates the probabilistic relationship between the stimulus (i.e. pixel values of a displayed RGB image) and the cone excitations (i.e. trial-by-trial photopigment isomerizations). ISETBio simulates the process of displaying an image on a monitor (*Figure 1A*), the wavelength-dependent optical blur of the human eye and spectral transmission through the lens and the macular pigment (*Figure 1B*), as well as the interleaved spatial and chromatic sampling of the retinal image by the L, M, and S cones (*Figure 1C*). Noise in the cone signals is characterized by a Poisson process. The forward model allows us to compute the *likelihood* function. The likelihood function represents the probability that an observed pattern of cone excitations was produced by any given image.

To obtain a *prior* over natural images, we applied independent components analysis (ICA, see Materials and methods) to a large dataset of natural images (*Russakovsky et al., 2015*), and fit an exponential probability density function to the individual component weights (*Figure 1D*). The prior serves as our description of the statistical structure of natural images.

Given the likelihood function, prior distribution, and an observed pattern of cone excitations, we can then obtain a reconstruction of the original image stimulus by applying Bayes rule to find the posterior probability of any image given that pattern. We take the reconstructed image as the one that maximizes the posteriori probability (MAP estimate, see Materials and methods) (*Figure 1D*).

### Basic properties of the reconstructions

To understand the consequences of initial visual encoding, we need to study the interaction between the likelihood function (i.e. our model of the initial encoding) and the statistics of natural images (i.e. the image prior). There are strong constraints on the statistical structure of natural images, such that natural images occupy only a small manifold within the space of all possible images. The properties of the initial encoding produce ambiguities with respect to what image is displayed when only the likelihood function is considered, but if these can be resolved by taking advantage of the statistical regularities of the visual environment, they should in principle, not prohibit effective visual perception. To illustrate this point, consider the simple example of discrete signal sampling: Based on the sampled signal, one cannot distinguish between the original signal from all its possible aliases (*Bracewell, 1986*). However, with the prior knowledge that the original signal contains only frequencies below the Nyquist frequency of the sampling array, this ambiguity is resolved. In the context of our current study, the role of the natural image prior comes in several forms, as we will demonstrate in Results. First, since the reconstruction problem is underdetermined, the prior is a regularizer, providing a unique MAP estimate; Second, the prior acts as a denoiser, counteracting the Poisson noise in the cone excitation; Lastly, the prior guides the spatial and spectral demosaicing of the signals provided via the discrete sampling of the retinal image by the cone mosaic.

To highlight the importance of prior information while holding the likelihood function fixed, we can vary a parameter $\gamma$ that adjusts the weight of the log-prior term in the reconstruction objective function (see Materials and methods). Explicitly manipulating $\gamma$ reveals the effect of the prior on the reconstruction (*Figure 2*). When $\gamma$ is small, the reconstruction is corrupted by the noise and the ambiguity of the initial visual encoding (*Figure 2A and B*). When $\gamma$ is large, the prior leads to desaturation and over-smoothing (*Figure 2E*) in the reconstruction. For the rest of our simulations, the value of $\gamma$ is determined on the training set by a cross-validation procedure that minimizes the reconstruction error, unless specified otherwise (*Figure 2C*).

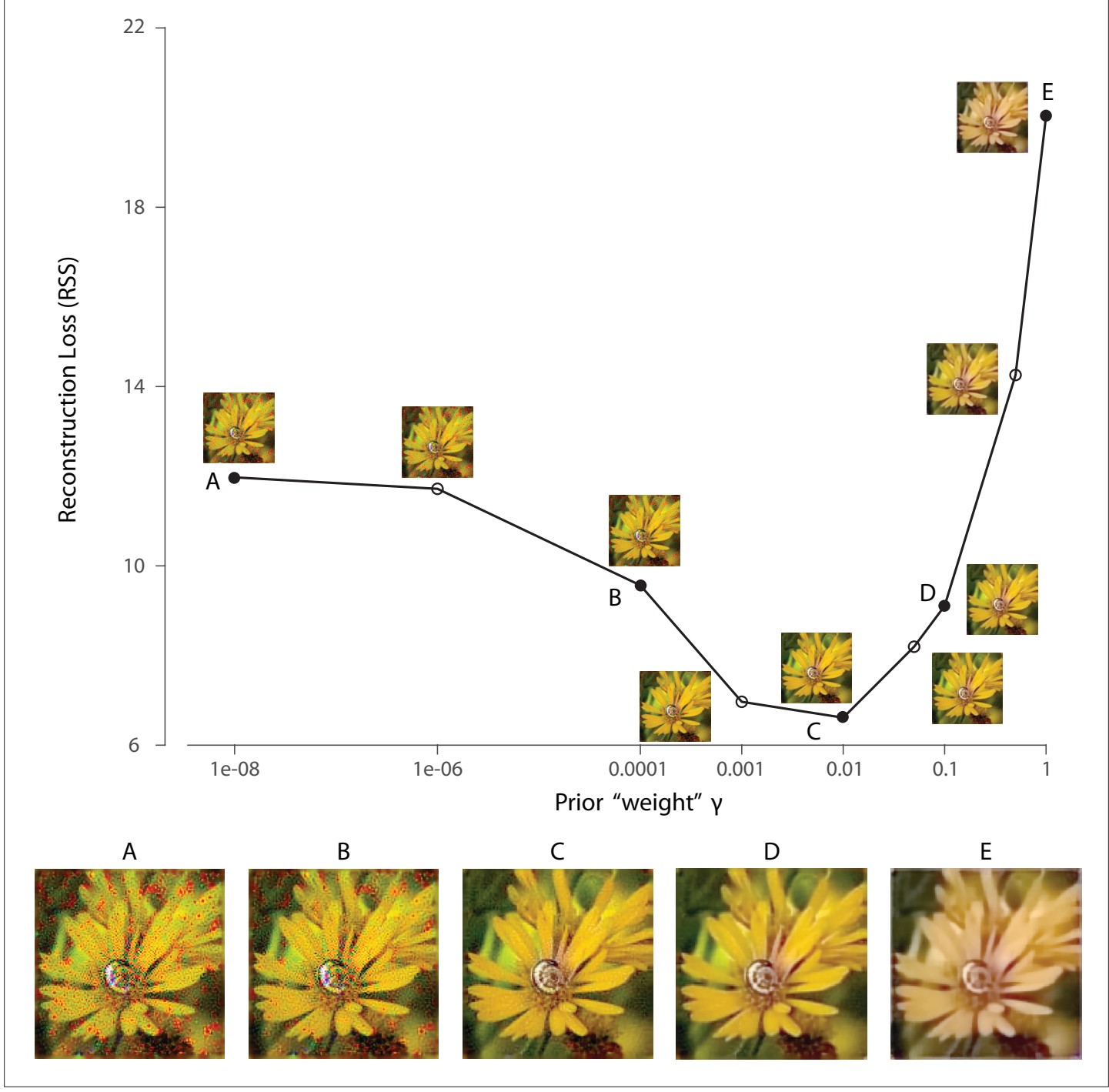

**Figure 2.** Effect of prior weight on reconstructed image. Reconstruction error for an example natural image using a 1 deg foveal mosaic and root sum of squared distance (RSS, y-axis) in the pixel space as the error metric, as a function of weight γ on the log-prior term (x-axis, see Materials and methods) in the reconstruction objective function. The reconstructed image obtained with each particular γ value is shown alongside each corresponding point. Image (C) corresponds to the value of γ obtained through the cross-validation procedure (see Materials and methods). The images at the bottom are magnified versions of a subset of the images for representative γ values, as indicated by the solid dots in the plot.

To further elucidate properties of the Bayesian reconstruction, especially the interaction between the likelihood and prior, we plotted a few representative images in a log-prior, log-likelihood coordinate system, given a particular instance of cone excitations (*Figure 3*). The optimal reconstruction, taken as the MAP estimate, has both a high prior probability and likelihood value as expected

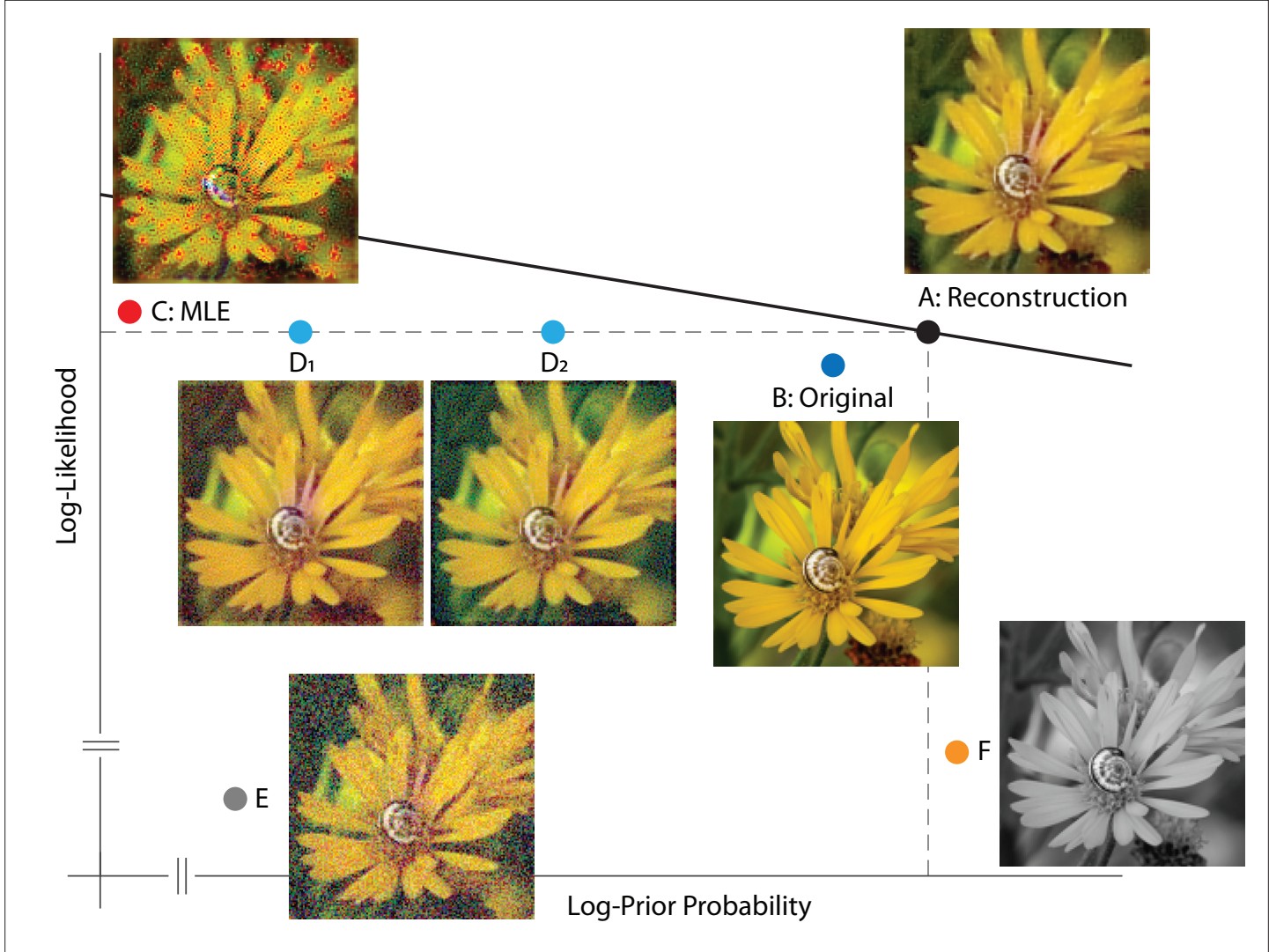

**Figure 3.** Solution space of image reconstruction. Given a particular instance of cone excitations, we can evaluate the (log-)prior probability (x-axis) and (log-)likelihood value (y-axis) for arbitrary images. Here, a few representative images are shown together with their corresponding location in a log-prior, log-likelihood coordinate system. (**A**) The optimal MAP reconstruction obtained via the reconstruction algorithm. The solid line shows $\gamma x + y = c$, with the value of $c$ evaluated at the optimal reconstruction and with the value of $\gamma$ matched to that obtained through cross-validation. (**B**) Original input image (ground truth). (**C**) A reconstruction generated by maximum likelihood estimation (MLE, set $\gamma = 0$). Note that the maximum likelihood reconstruction shown is not unique, since adding any pattern from the null space of the likelihood matrix leads to a different reconstruction with the same maximum likelihood. Here one arbitrarily chosen MLE reconstruction is shown. (**D**) Optimal reconstruction, corrupted by patterns randomly sampled from the null space of the likelihood render matrix (see Materials and methods). These have the same likelihood as the optimal reconstruction, but lower prior probability. (**E**) Optimal reconstruction, corrupted by white noise in RGB space. (**F**) Grayscale version of the optimal reconstruction.

(*Figure 3A*). In fact, for our reconstruction algorithm, there should not exist any image above the $\gamma x + y = c$ line that goes through **A** (solid line, *Figure 3*), otherwise the optimization routine has failed to find the global optimum. The original image stimulus (ground truth) has a slightly lower likelihood value, mainly due to noise present in the cone excitations, and also a slightly lower prior probability, possibly due to the fact that our prior is only an approximation to the true natural image distribution (*Figure 3B*). The detrimental effect of noise becomes prominent in a maximum likelihood estimate (MLE, *Figure 3C*): Noise in the cone excitations is interpreted as true variation in the original image stimulus, thus slightly increasing the likelihood value but also creating artifacts. Such artifacts are penalized by the prior in other reconstructions. Furthermore, even without the presence of noise, other features of the initial visual encoding (e.g. *Figure 1B and C*) cause loss of information and ambiguity for the reconstruction. This is illustrated by a set of images that lie on the equal likelihood line

with the MAP reconstruction (*Figure 3D*): There exist an infinite set of variations in the image (stimulus) that have no effect on the value of the likelihood function (i.e. variations within the null space of the linear likelihood render matrix, see Materials and methods). Thus, the cone excitations provide no information to distinguish between images that differ by such variations. However, as with the case of noise, variations inconsistent with natural images are discouraged by the prior. (Another implication of the existence of the null space is that the MLE solution to the reconstruction problem is actually underdetermined, as an entire subspace of images can have the same likelihood value. In the figure we show one arbitrarily chosen MLE estimate.) Other corruptions of the image, such as addition of white noise in the RGB pixel space, are countered by both the likelihood and prior (*Figure 3E*). Lastly, for illustrative purposes, we can increase the prior probability of the reconstruction relative to the optimal by making it spatially or chromatically more uniform (*Figure 3F*), but doing so decreases the likelihood.

## Optimal allocation of retinal photoreceptors

Within the Bayesian reconstruction framework, the goal of the visual front end can be characterized as minimizing the average error in reconstruction across the set of natural images. In this context, we can ask how to choose various elements of the initial encoding, subject to constraints, to minimize the expected reconstruction error under the natural image prior (*Levin et al., 2008*; *Manning and Brainard, 2009*). More formally, we seek the 'design' parameters $\theta$ of a visual system:

$$\theta = \underset{\theta}{\text{argmin}} \; E_{p(x)} \left[ E_{p(m|x; \theta)} \left[ L(\hat{x}(m; \theta), \, x) \right] \right],$$

where $\hat{x}(m; \theta) = \text{argmax}_x \; p(m|x; \theta) \, p(x)$. Here, $x$ represents individual samples of natural images, $m$ represents instances of cone excitation (i.e. sensory measurements), and $p(m|x; \theta)$ is our model of the initial encoding (i.e. likelihood function). The particular features under consideration of the modeled visual system are indicated explicitly by the parameter vector $\theta$. The MAP image reconstruction is indicated by $\hat{x}(m; \theta)$, and $L(\cdot, \cdot)$ is a loss function that assesses reconstruction error. In practice, the expectations are approximated by taking the average over large samples of natural images and cone excitations. (For simplicity in the development here, we did not include the parameter $\gamma$ that we incorporated into our reconstruction algorithm in the equations above. It was included in the actual computations that investigated the reconstruction performance. Also note that the MAP estimate is not in general the one that minimizes the expected loss. We use the MAP estimate as a computationally tractable proxy for the loss-minimizing estimate.)

One intriguing design problem is the allocation of cone photoreceptor types: The maximum number of photoreceptors (cones) per unit area is bounded due to biological constraints. How should the visual system assign this limited resource across the three different types of cones? It has been observed in human subjects that there is a relatively sparse population of S cones, while large individual variability exists in the L/M cone ratio (*Hofer et al., 2005*). Previous research has used information-theoretical measures combined with approximations to address this question (*Garrigan et al., 2010*). Here, we empirically evaluated a loss function (i.e. we used root sum of squares distance in the RGB pixel space as well as the S-CIELAB space) on the reconstructed images, while systematically changing the allocation of retinal cone types (*Figure 4*).

Interestingly, we found that large variations (nearly a 10-fold range) in the assignment of L and M cones have little impact on the average reconstruction error (*Figure 4A*). Only when the proportion of L or M cones becomes very low is there a substantial increase in reconstruction error, as the modeled visual system approaches dichromacy. On the other hand, the average reconstruction error as a function of the proportion of S cones shows a clear optimum at a small S-cone proportion (~10%; *Figure 4B*).

Our results are in agreement with a previous analysis in showing that the empirically observed allocation of retinal photoreceptor type is consistent with the principle of optimal design (*Garrigan et al., 2010*; also see *Levin et al., 2008*; *Manning and Brainard, 2009*; *Sampat et al., 2015*; *Jiang et al., 2017*). The indifference to L/M ratio can be explained by the large spatial and chromatic correlations present in natural images, together with the high overlap in L- and M-cone spectral sensitivities. This leads to a high correlation in the excitations of neighboring L and M cones in response to natural images, allowing cones of one type to be substituted for cones of the other type with little effect on reconstruction error (see the next paragraph for additional analysis on this point). Additional analysis

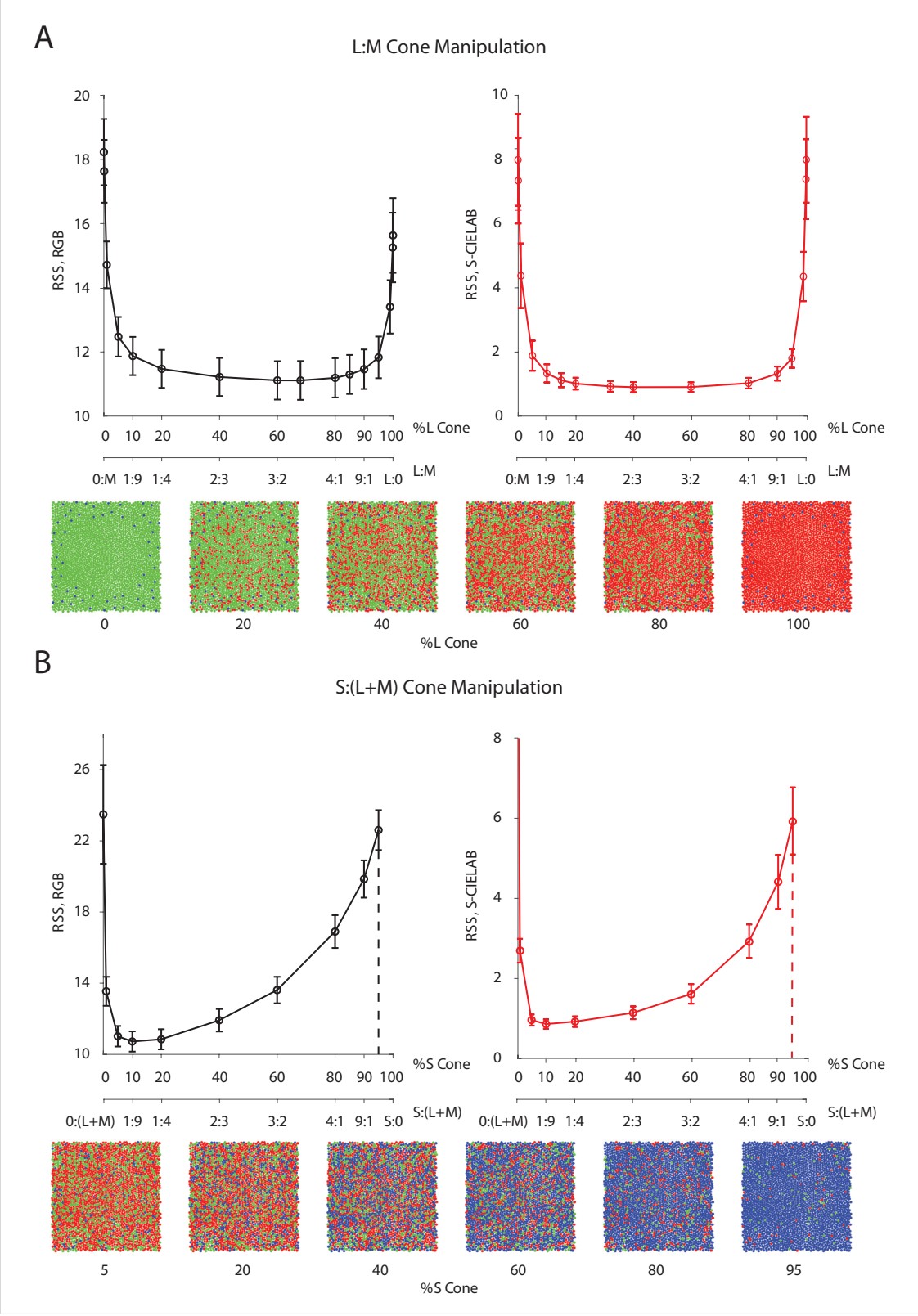

**Figure 4.** Effect of the allocation of retinal cone types on reconstruction. Average image reconstruction error from a 1 deg foveal mosaic on a set of natural images from the evaluation set, computed as root sum of squares (RSS) distance in the RGB pixel space (y-axis, left panels) and the S-CIELAB space (y-axis, right panels), as a function of different allocations of retinal photoreceptor (cone) types in the mosaic. (**A**) Average (over evaluation images) reconstruction error as a function of %L cone (top x-axis), or L:M cone ratio (bottom x-axis). Example mosaics with different %L values are

*Figure 4 continued on next page*

*Figure 4 continued*

shown below the plot. Error bars indicate ±1 SEM. (**B**) Average reconstruction error as a function of %S cone (top x-axis), or S:(L + M) cone ratio (bottom x-axis). Example mosaics with different %S values are shown below the plot. Error bars indicate ±1 SEM across sampled images. See *Figure 4—figure supplement 2* for a replication of the same analysis with hyperspectral images.

The online version of this article includes the following figure supplement(s) for figure 4:

**Figure supplement 1.** Factors that contribute to optimal S cone proportion.

**Figure supplement 2.** Effect of the allocation of retinal cone types on reconstruction of hyperspectral images.

(*Figure 4—figure supplement 1*) revealed that the sensitivity to S cone proportion is due to a combination of two main factors: (1) chromatic aberrations, which blur the retinal image at short wavelengths and reduce the value of dense spatial sampling at these wavelengths; and (2) S cones mainly contribute to the estimation of pixel values in the B-pixel plane, whereas L and M cone contribute to both the R- and G-pixel planes (see *Figure 4—figure supplement 1*). This makes L and M cones more informative than S cones, given the particular loss functions we employ to evaluate reconstruction error. To further validate our conclusion, we have also replicated our analysis with a dataset of hyperspectral (as opposed to RGB) images (*Nascimento et al., 2002*; *Chakrabarti and Zickler, 2011*), with a loss function applied directly to the whole spectrum, and have obtained similar results (*Figure 4—figure supplement 2*, also see Materials and methods).

To further study the role of statistical regularities in the optimal allocation of photoreceptor type, we repeated the L-cone proportion analysis above, but on different sets of synthetic image datasets for which the spatial and chromatic correlations in the images were manipulated explicitly (see Materials and methods). The dependence of the average reconstruction error on the L-cone proportion decreases as the chromatic correlation in the signal increases (*Figure 5*). A decrease of spatial correlation has little impact on the shape of the curves, but increases the overall magnitude of reconstruction error (*Figure 5*; to highlight the shape, the scale of the y-axis is different across rows and columns. See *Figure 5—figure supplement 1* for the same plot with matched y-axis scale). When both the chromatic and spatial correlation are high, there is a large margin of L-cone proportion within which the reconstruction error is close to the optimal (minimal) point (*Figure 5*, shaded area). This analysis highlights the importance of considering visual system design in the context of the statistical properties (prior distribution) of natural images, as it shows that the conclusions drawn can vary with these properties (*Barlow, 1961*; *Derrico and Buchsbaum, 1991*; *Barlow and Földiàgk, 1989*; *Atick et al., 1992*; *Lewis and Zhaoping, 2006*; *Levin et al., 2008*; *Borghuis et al., 2008*; *Garrigan et al., 2010*; *Tkacik et al., 2010*; *Atick, 2011*; *Burge, 2020*). Natural images are thought to have both high spatial and high chromatic correlation (*Webster and Mollon, 1997*; *Nascimento et al., 2002*; *Garrigan et al., 2010*), making the results shown in *Figure 5* consistent with those in *Figure 4*.

## Visualization of color deficiency with image reconstruction

In addition to quantification, the reconstruction framework also provides a method for visualizing the effect of information loss in the initial visual encoding. We know that extreme values of L:M cone ratio create essentially dichromatic retinal mosaics, and from the analysis above we observed that these lead to high reconstruction error. To understand the nature of this error, we can directly visualize the reconstructed images.

*Figure 6A* shows reconstructions of a set of example images from different dichromatic retinal mosaics. While the spatial structure of the original images is largely retained in the reconstructions, each type of dichromacy creates a distinct pattern of color confusions and shifts in the reconstructed color. Note that in the case where there is no simulated cone noise (as in *Figure 6*), the original image has a likelihood at least as high as the reconstruction obtained via our method. Thus, the difference between the original images and each of the corresponding dichromatic reconstructions is driven by the image prior. On the other hand, the difference in the reconstructions across the three types of dichromacy illustrates how the different dichromatic likelihood functions interact with the prior.

One might speculate as to whether the reconstructions predict color appearance as experienced by dichromats. To approach this, we compare the reconstructions with two other methods that have been proposed to predict the color appearance for dichromats (*Brettel et al., 1997*; *Jiang et al., 2016*). To determine an image based on the excitations of only two classes of cones, any method

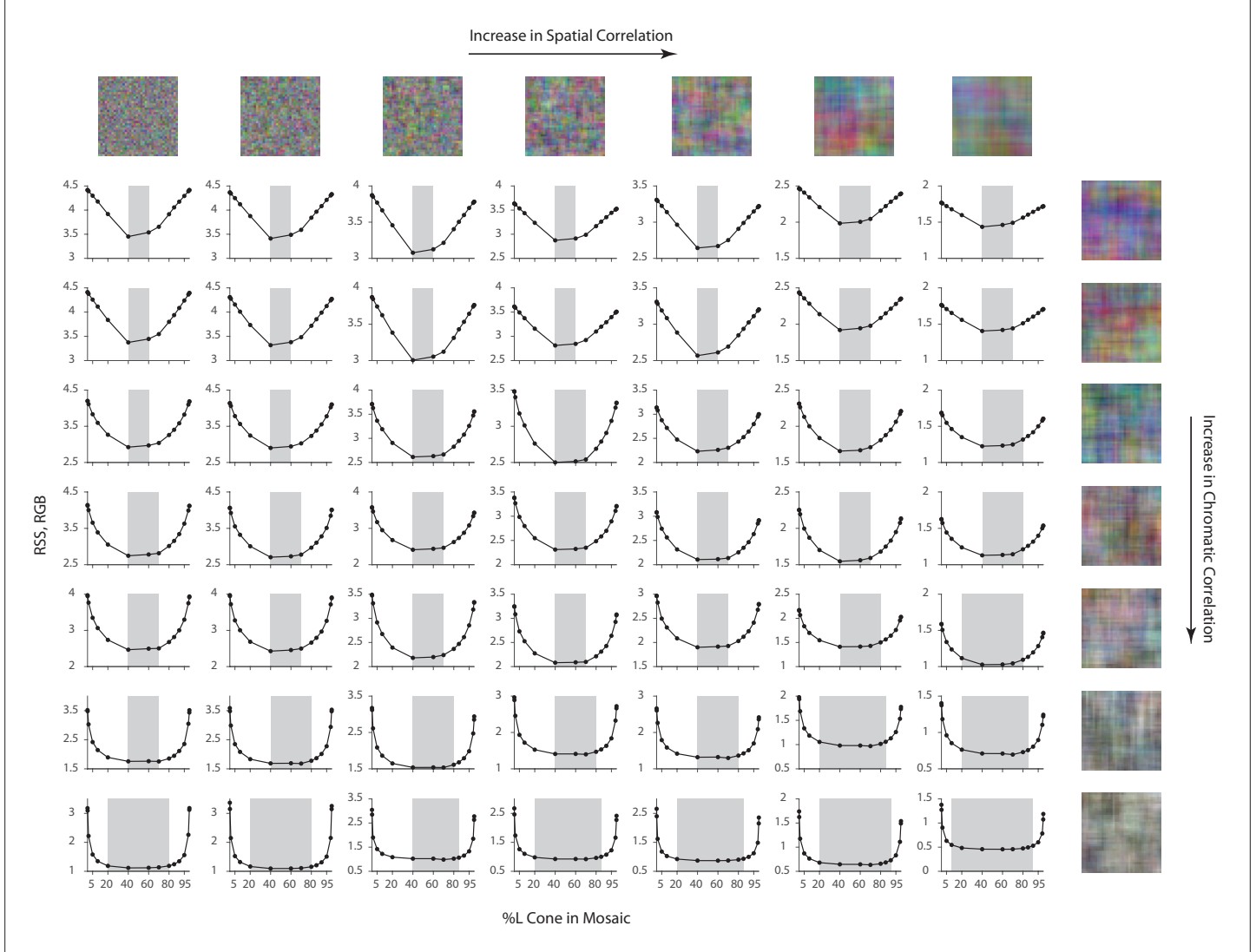

**Figure 5.** Effect of spatial and chromatic correlation on the optimal allocation of photoreceptors. Average image reconstruction error from a half-degree square foveal mosaic on different sets of synthetic images, computed as root sum of squares (RSS) distance in the RGB pixel space, as a function of %L cone (L:M cone ratio) of the mosaic (i.e. similar to **Figure 4A**, left column). The shaded areas represent %L values that correspond to RSS values within a +0.1 RSS margin of the optimal (minimum RSS) point. Within each panel, synthetic images were sampled from a Gaussian distribution with specified spatial and chromatic correlation, as indicated by example images on the top row and rightmost column, and reconstruction was performed with the corresponding Gaussian prior (see Materials and methods). The overall RSS is reduced compared to **Figure 4** due to the smaller image size used and the fact that the images were drawn from a different prior, as well as because the prior used in reconstruction exactly describes the images for this case. In addition, reconstruction error bars are negligible due to the large image sample size used.

The online version of this article includes the following figure supplement(s) for figure 5:

**Figure supplement 1.** Effect of spatial and chromatic correlation on the optimal allocation of photoreceptors (with matched y-axis).

will need to rely on a set of regularizing assumptions to resolve the ambiguity introduced by the dichromatic retinas. *Brettel et al., 1997* started with the trichromatic cone excitations of each image pixel, and projected these onto a biplanar surface, with each plane defined by the neutral color axis and an anchoring stimulus identified through color appearance judgments made across the two eyes of unilateral dichromats. The resulting trichromatic excitations were then used to determine the rendered RGB values (*Figure 6B*). *Jiang et al., 2016* also adopted a reconstruction approach, but one that reconstructed the incident spectrum from the dichromatic cone excitations at each pixel. They then projected the estimated spectra onto trichromatic cone excitations, and used these to render the RGB values (*Figure 6C*). In their method, a spectral smoothness constraint was introduced to

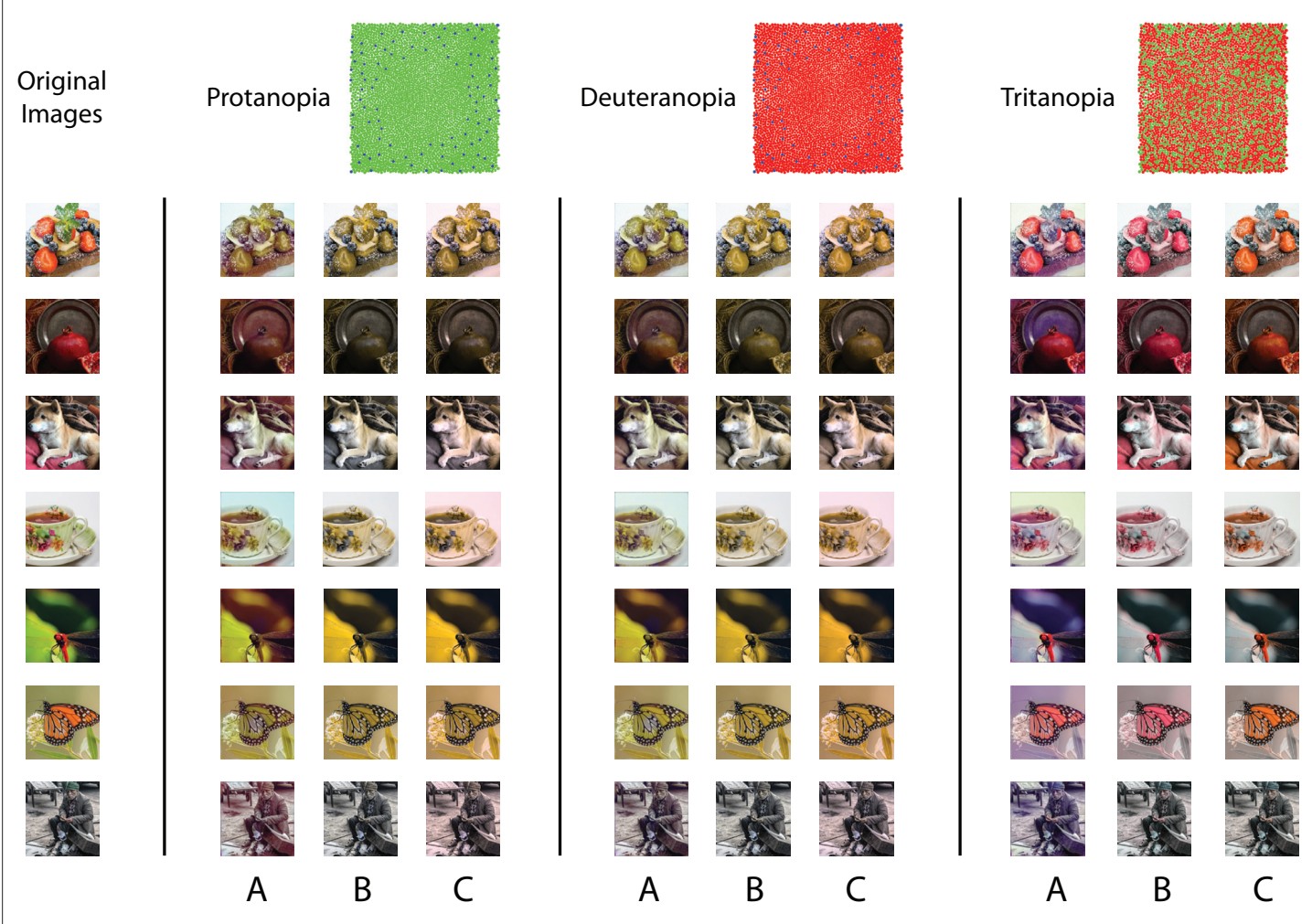

**Figure 6.** Visualization of the effect of dichromacy. Reconstructions of a set of example images in the evaluation set from different types of 1 degree foveal dichromatic retinal mosaics (protanopia, deuteranopia, tritanopia) together with other previously proposed methods for predicting color appearance for dichromats. (**A**) Our method; (**B**) *Brettel et al., 1997*; (**C**) *Jiang et al., 2016*. Cone noise was not simulated for the images shown in this figure, since the comparison methods operate directly on the input images. See Materials and methods for a brief description of the implementation of the two other methods.

regularize the spectral estimates, which favors desaturated spectra. In this sense, their prior is similar to ours: The sparse prior we used is centered on the average image, which is desaturated, and also encourages achromatic content due to the high correlations across color channels. One noticeable difference between our method and the other two is that ours takes into account the spatial structure of the image.

Interestingly, although there are differences in detail between the images obtained, in many cases the different methods produce visualizations that are quite similar. We find the general agreement between the reconstruction-based methods and the one based on subject reports an encouraging sign that the reconstruction approach can be used to predict aspects of appearance.

Anomalous trichromacy is another form of color deficiency that is commonly found in human observers. For example, in deuteranomaly, the spectral sensitivity of the M cones is shifted toward that of the L cones (*Figure 7B*). Since the three cone spectral sensitivity functions are linearly independent of each other, in the absence of noise we should be able to obtain a trichromatic reconstruction from the excitations of the deuteranomalous mosaic. However, in the presence of noise, we expect that the high degree of overlap between M and L spectral sensitivities will result in a lower signal-to-noise ratio (SNR) in the difference between M- and L-cone excitations, compared to that of a normal trichromatic observer, and thus lead to worse reconstructions. We performed image reconstructions

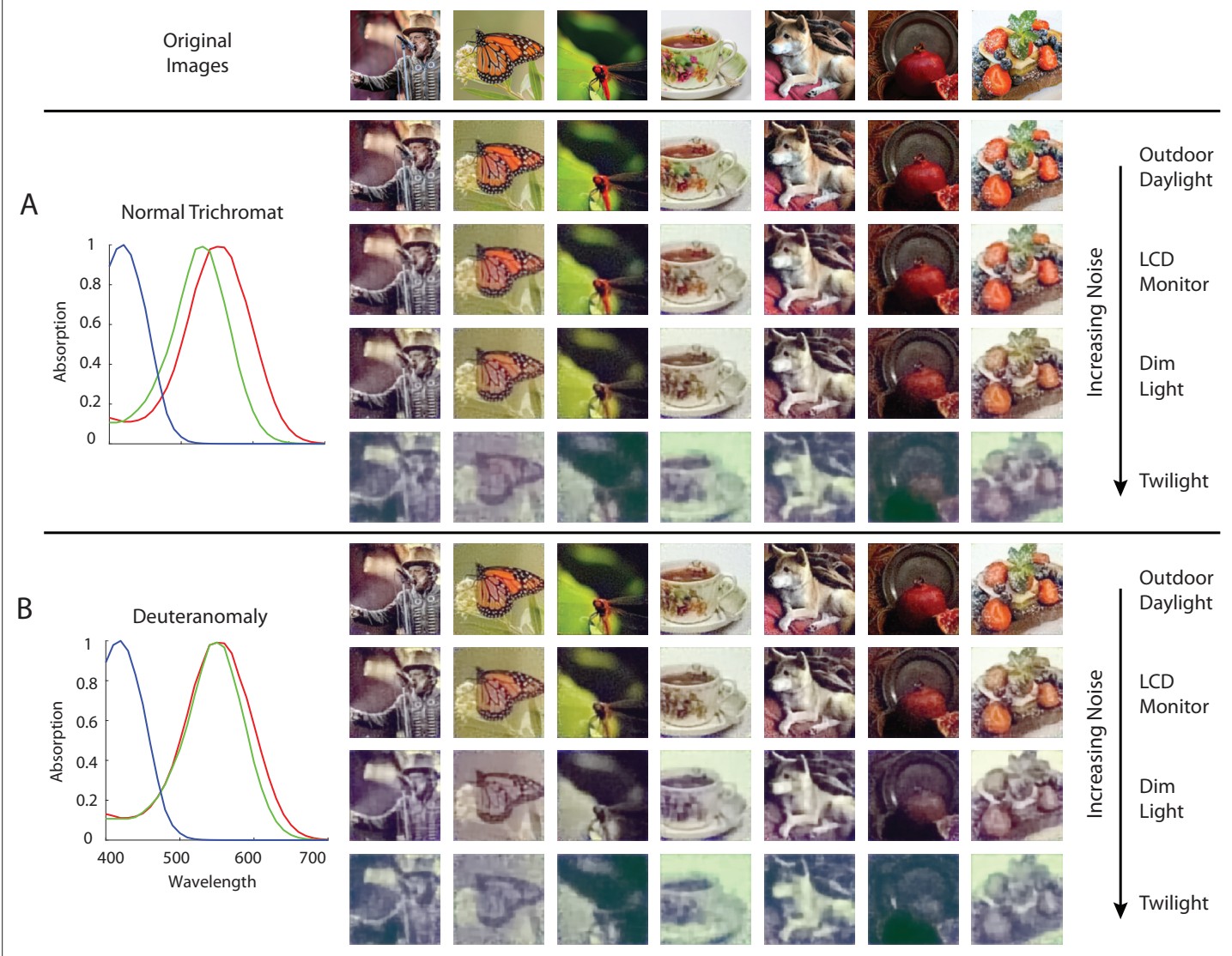

**Figure 7.** Comparison of normal and deuteranomalous observers at varying light intensities. Image reconstructions for a set of example images in the evaluation set from 1 degree, foveal (**A**) normal trichromatic and (**B**) deuteranomalous trichromatic mosaics at four different overall light intensity levels that lead to different Poisson signal-to-noise ratios in the cone excitations. The average excitations (photo-isomerizations) per cone per 50ms integration time is chosen to be approximately $10^4$ for *Outdoor Daylight*, $10^3$ for *LCD Monitor*, $10^2$ for *Dim Light*, and $10^1$ for *Twilight* (***Lewis and Zhaoping, 2006***; ***Stockman and Sharpe, 2006***). The prior weight parameter in these set of simulations was set based on a cross-validation procedure that minimizes RMSE $\left(\lambda = 0.05\right)$. To highlight interaction between noise and the prior, we have also included a set of reconstructions with the prior weight set to a much lower level $\left(\lambda = 0.001\right)$, see ***Figure 7—figure supplement 1***.

The online version of this article includes the following figure supplement(s) for figure 7:

**Figure supplement 1.** Reconstruction with a weak prior across SNR levels.

for a normal trichromatic (with a peak spectral sensitivity of M cone at 530 nm) and a deuteranomalous (with a peak spectral sensitivity of M cone at 550 nm) 1 deg foveal mosaic at different overall light intensity levels (***Figure 7***). Due to the nature of Poisson noise, the higher the light intensity, the higher the SNR of the cone excitations. At high light intensities, the reconstructions are similar for the normal and deuteranomalous mosaics (first row). At lower intensities, however, the deuteranomalous reconstruction lacks chromatic content still present in the normal reconstruction (second and third row). The increase in noise also reduces the amount of spatial detail in the reconstructed images, due to the denoising effect driven by the image prior. Furthermore, a loss of chromatic content is also seen for the reconstruction from the normal mosaic at the lowest light level (last row). This observation may be

connected to the fact that biological visual systems that operate at low light levels are typically mono-chromatic, potentially to increase the SNR of spatial vision at the cost of completely disregarding color (e.g. the monochromatic human rod system; see *Manning and Brainard, 2009* for a related and more detailed treatment; also see *Walls, 1942*; *Rushton, 1962*; *van Hateren, 1993*; *Land and Osorio, 2003*).

## Effect of physiological optics and mosaic spatial sampling

So far, our visualizations have focused on chromatic information loss due to a reduced number of cone types or a shift in cone spectral sensitivity. However, imperfection in the physiological optics, combined with the spatial sampling of retinal mosaic, also introduces significant loss of information. Furthermore, the interleaved nature of the mosaic means that color and pattern are entangled at the

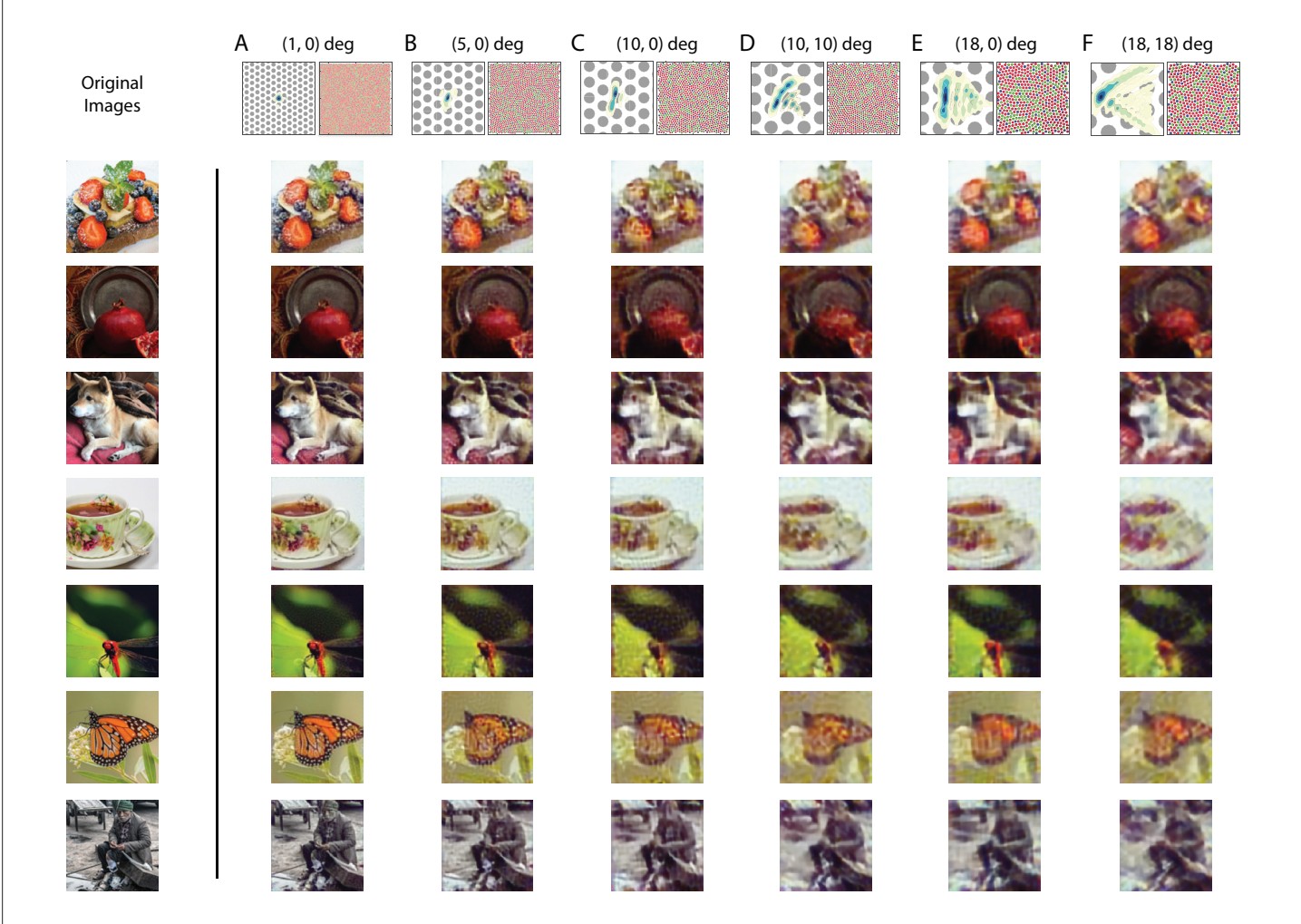

**Figure 8.** Image reconstruction with optics/mosaic at different retinal eccentricities. Image reconstructions for a set of example images in the evaluation set from 1 degree patches of mosaic at different retinal eccentricities. The coordinates at the top of each column indicate the horizontal and vertical eccentricity of the patch used for that column. The image at the top left of each column shows a contour plot of the point-spread function relative to an expanded view of the cone mosaic used for that column, while the image at the top right of each column shows the full 1 degree mosaic (see *Figure 8—figure supplement 1* for an enlarged view of the mosaic and optics).

The online version of this article includes the following figure supplement(s) for figure 8:

**Figure supplement 1.** Optics and cone mosaic at different retinal eccentricities.

**Figure supplement 2.** Reconstruction error at different visual eccentricities.

**Figure supplement 3.** Image reconstruction with different point spread functions.

**Figure supplement 4.** Image reconstruction at peripheral eccentricities with maximum likelihood estimation (MLE).

very initial stage of visual processing (*Brainard, 2019*). To highlight these effects, we reconstructed natural images from 1 deg patches of mosaics at different retinal eccentricities across the visual field, with (1) changes in optical aberrations (*Polans et al., 2015*); (2) increases in size and decreases in density of the photoreceptors (*Curcio et al., 1990*); and (3) decreases in the density of the macular pigment (*Nolan et al., 2008*; *Putnam and Bland, 2014*). The degradation in the quality of the reconstructed images can be clearly observed as we move from the fovea to the periphery (*Figure 8*; See *Figure 8—figure supplement 1* for an enlarged view of the mosaic and optics). For some retinal locations, the elongated point-spread function (PSF) also introduces a salient directional blur (*Figure 8E and F*). For a simple quantification of the average reconstruction error as a function of visual eccentricity, see *Figure 8—figure supplement 2*.

The consequences of irregular spatial sampling by the cone mosaic have been previously studied with the framework of signal processing (*Snyder et al., 1977*; *Yellott, 1983*). Our results highlight that optimizing the initial visual encoding depends in rich ways on the interplay between the cone sampling and the optics. While less information (i.e. at more eccentric locations) does lead to overall lower quality reconstructions (*Figure 8—figure supplement 2*), exactly which aspects of the reconstructions are incorrect can vary in subtle ways. Concretely, in *Figure 8*, we observe a trade-off across visual eccentricity between spatial and chromatic vision. In the image of the dragonfly, for example, the reconstructed colors are desaturated at intermediate eccentricities (e.g. *Figure 8C and D*), compared with the fovea (*Figure 8A*) and more eccentric locations (*Figure 8E and F*). The desaturation is qualitatively consistent with the literature that indicates a decrease in chromatic sensitivity at peripheral visual eccentricities, at least for the red-green axis of color perception and for some stimulus spatial configurations (*Virsu and Rovamo, 1979*; *Mullen and Kingdom, 1996*; but see *Hansen et al., 2009*). To further elucidate this richness, in an additional analysis, we systematically varied the size of the PSF for a fixed peripheral retinal mosaic. This revealed that (*Figure 8—figure supplement 3*): (1) A larger PSF does lead to better estimate of chromatic content, albeit eventually at the cost of spatial content. (2) In general, an appropriate amount of optical blur is required to achieve the best overall image reconstruction performance, presumably due to its prevention of aliasing. We will treat the issue of spatial aliasing further in the next section.

Lastly, to emphasize the importance of the natural image prior, we performed a set of maximum likelihood reconstructions with no explicit prior constraint, which resulted in images with less coherent spatial structure and lower fidelity color appearance (*Figure 8—figure supplement 4*). Thus, the prior here is critical for the proper demosaicing and interpolation of the information provided by the sparse cone sampling at these peripheral locations.

## Spatial aliasing

As we have alluded to above, the retinal mosaic and physiological optics can also interact in other important ways: Both in humans and other species, it has been noted that the optical cut-off of the eye is reasonably matched to the spacing of the photoreceptors (i.e. the mosaic Nyquist frequency), enabling good spatial resolution while minimizing spatial aliasing due to discrete sampling (*Williams, 1985*; *Snyder et al., 1986*; *Land and Nilsson, 2012*). In contrast to our work, these analyses did not take into account the fact that the cone mosaic interleaves multiple spectral classes of cones (but see *Williams et al., 1991*; *Brainard, 2015*), and here we revisit classic experiments on spatial aliasing for a trichromatic mosaic using our reconstruction framework.

Experimentally, it has been demonstrated that with instruments that *bypass* the physiological optics and present high contrast grating stimuli directly on the retina, human subjects can detect spatial frequencies up to 200 cyc/deg (*Williams, 1985*). For foveal viewing, subjects also report having a percept resembling a pattern of 'two-dimensional noise' and/or 'zebra stripes' when viewing those high spatial frequency stimuli (*Williams, 1985*). For peripheral viewing, high frequency vertical gratings can be perceived as horizontal (and vice-versa; *Coletta and Williams, 1987*). We explored these effects within our framework as follows: We reconstructed a set of vertical chromatic grating stimuli from the cone excitations of a foveal and a peripheral mosaic. To simulate the interferometric experimental conditions of *Williams, 1985*, we used diffraction-limited optics with no longitudinal chromatic aberration (LCA), allowing high-frequency stimuli to reach the cone mosaic directly. For gratings that are above the typical optical cut-off frequency, we obtained reconstructions that (1) are quite distinct from a uniform field, which would allow them to be reliably detected in a discrimination

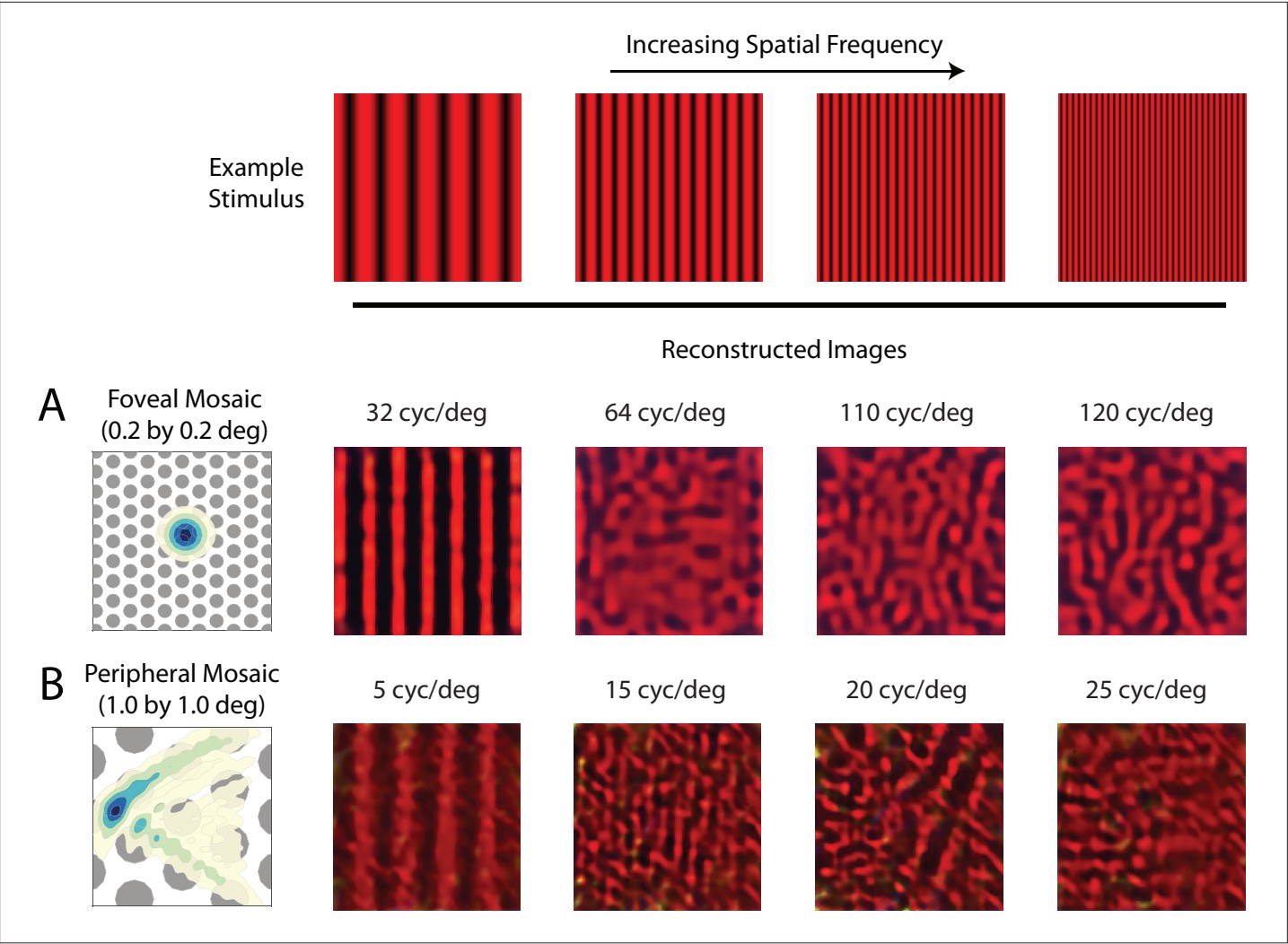

**Figure 9.** Reconstruction of chromatic grating stimuli without optical aberrations. Image reconstruction of chromatic grating stimuli with increasing spatial frequency from (**A**) a 0.2 deg foveal mosaic and (**B**) a 1 deg peripheral mosaic at (18, 18) degree retinal eccentricity, using diffraction-limited optics without LCA. The leftmost images show an expanded view of the cone mosaic relative to a contour plot of a typical point-spread function at that eccentricity. Images were modulations of the red channel of the simulated monitor, to mimic the 633 nm laser used in the interferometric experiments. The exact frequency of the stimuli being used for each condition is as denoted in the figure. For a more extended comparison between reconstructions with and without optical aberrations, see *Figure 9—figure supplement 1* and *Figure 9—figure supplement 2*.

The online version of this article includes the following figure supplement(s) for figure 9:

**Figure supplement 1.** Reconstruction of chromatic grating stimuli with/without optical aberrations.

**Figure supplement 2.** Reconstruction of achromatic grating stimuli with/without optical aberrations.

protocol; and (2) lack the coherent vertical structure of the original stimulus (*Figure 9*). Concretely, the reconstructions recapitulate the 'zebra stripe' percept reported at approximately 120 cyc/deg in the fovea (*Figure 9A*); as well as the orientation-reversal effect at an appropriate spatial frequency in the periphery (*Figure 9B*). Both results corroborate previous theoretical analysis and psychophysical measurements (*Williams, 1985*; *Coletta and Williams, 1987*), but now taking the trichromatic nature of the mosaic into account. On the other hand, with full optical aberrations, the reconstructed images became mostly uniform at these high spatial frequencies (*Figure 9—figure supplement 1*). Since our method accounts for trichromacy, we have also made the prediction that for achromatic grating stimuli viewed under similar diffraction-limited conditions, while the spatial aliasing pattern will be comparable, additional chromatic aliasing should be visible (*Figure 9—figure supplement 2*; also see *Williams et al., 1991*; *Brainard, 2015*).

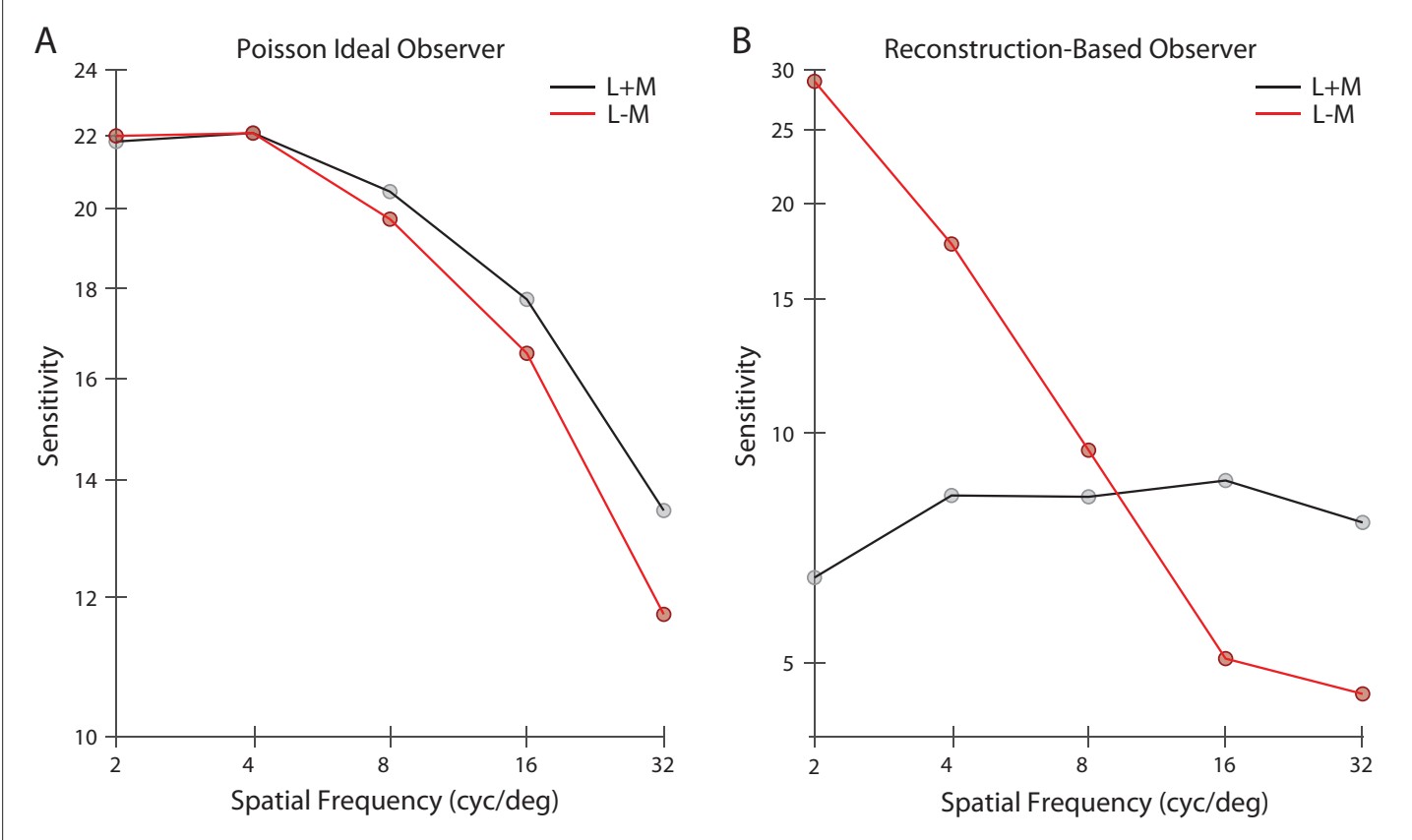

**Figure 10.** Contrast sensitivity functions. Contrast sensitivity, defined as the inverse of threshold contrast, for (**A**) a Poisson 2AFC ideal observer, and (**B**) an image reconstruction-based observer (see Materials and methods), as a function of the spatial frequency of stimulus in either the L + M direction (black) and L - M cone contrast direction (red). Contrast was measured as the vector length of the cone contrast vector, which is matched across the two color directions.

The online version of this article includes the following figure supplement(s) for figure 10:

**Figure supplement 1.** Contrast sensitivity function of a MLE reconstruction observer.

## Contrast sensitivity function

Our framework can also be adapted to perform ideal observer analysis for psychophysical discrimination (threshold) tasks, which have been used previously to evaluate the information available in the initial encoding. Here, we use the reconstructed images as the basis for discrimination decisions. This is potentially important since even the early post-receptoral visual representation (e.g. retinal ganglion cells), on which downstream decisions must be based, is likely shaped by the regularities of our visual environment (*Atick et al., 1992*; *Borghuis et al., 2008*; *Karklin and Simoncelli, 2011*; *Atick, 2011*). Our method provides a way to extend ideal observer analysis to incorporate these statistical regularities.

Concretely, we predicted and compared the diffraction-limited spatial contrast sensitivity function (CSF) for gratings with a half-degree spatial extent (see Materials and methods). First, we applied the classic signal-known-exactly ideal observer to the Poisson distributed excitations of the simulated cone mosaic. We computed CSFs for both achromatic (L + M) and chromatic (L - M) grating modulations, with matched cone contrast measured as the vector length of the cone contrast vector. As expected, the ideal observer at the cone excitations produces nearly identical CSFs for the contrast-matched L + M and L - M modulations; also, as expected, these fall off with spatial frequency, primarily because of optical blur (*Figure 10A*).

Next, we reconstructed images from the cone excitations produced by the grating stimuli. A template-matching observer based on the noise-free reconstructions was then applied to the noisy reconstructions (see Materials and methods). The image-reconstruction observer shows significant

interactions between spatial frequency and chromatic direction. Sensitivity in the L + M direction is relatively constant with spatial frequency. Sensitivity in the L − M direction starts out higher than L + M at low spatial frequencies, but drops significantly and is lower than L + M at high spatial frequencies (*Figure 10B*). We attribute these effects to the role of the image prior in the reconstructions, which leads to selective enhancement/attenuation of different image components. In support of this idea, we also found that an observer based on maximum likelihood reconstruction without the explicit prior term produced CSFs similar in shape to the Poisson ideal observer (*Figure 10—figure supplement 1*).

It is intriguing that the CSFs from the reconstruction-based observer show substantially higher sensitivity for L - M than for L + M modulations at low spatial frequencies (with equated RMS cone contrast), but with a more rapid falloff such that the sensitivity for L + M modulations is higher at high spatial frequencies. Both of these features are characteristic of the CSFs of human vision (*Mullen, 1985*; *Anderson et al., 1991*; *Chaparro et al., 1993*; *Sekiguchi et al., 1993*). A more comprehensive exploration of this effect and its potential interaction with other decision rule used in the calculation awaits future research.

## Discussion

We developed a Bayesian image reconstruction framework for characterizing the initial visual encoding, by combining an accurate image-computable forward model together with a sparse coding model of natural image statistics. Our method enables both quantification and visualization of information loss due to various factors in the initial encoding, and unifies the treatment of a diverse set of issues that have been studied in separate, albeit related, ways. In several cases, we were able to extend previous studies by eliminating simplifying assumptions (e.g. by the use of realistic, large cone mosaics that operate on high-dimensional, naturalistic image input). To summarize succinctly, we highlight here the following novel results and substantial extensions of previous findings: (1) When considering the allocation of different cone types on the human retina, we demonstrated the importance of the spatial and spectral correlation structure of the image prior; (2) As we examined reconstructions as a way to visualize information loss, we observed rich interactions in how the appearances of the reconstruction vary with mosaic sampling, physiological optics, and the SNR of the cone excitations; (3) We found that the reconstructions are consistent with empirical reports of retinal spatial aliasing obtained with interferometric stimuli, adding an explicit image prior component and extending consideration of the interleaved nature of the trichromatic retinal cone mosaic relative to the previous treatment of these phenomena; (4) We linked image reconstructions to spatio-chromatic contrast sensitivity functions by applying a computational observer for psychophysical discrimination to the reconstructions. Below, we provide an extended discussion of key findings, as well as of some interesting open questions and future directions.

First, we cast retinal mosaic design as a 'likelihood design' problem. We found that the large natural variations of L- and M-cone proportion, and the relatively stable but small S-cone proportion, can both be explained as an optimal design that minimizes the expected image reconstruction loss. This is closely related to an alternative formalism, often termed 'efficient coding', which seeks to maximize the amount of information transmission (*Barlow, 1961*; *Karklin and Simoncelli, 2011*; *Wei and Stocker, 2015*; *Sims, 2018*). In both cases, the optimization problem is subject to realistic biological constraints and incorporates natural scene statistics. Previous work (*Garrigan et al., 2010*) conducted a similar analysis with consideration of natural scene statistics, physiological optics, and cone spectral sensitivity, using an information maximization criterion. One advance enabled by our work is that we are able to fully simulate a 1 deg mosaic with naturalistic input, as opposed to the information-theoretical measures used by Garrigan et al., which became intractable as the size of the mosaic and the dimensionality of the input increased. In fact, *Garrigan et al., 2010* approximated by estimating the exact mutual information for small mosaic size ($N = 1 \ldots 6$ cones) and then extrapolated to larger cone mosaics using a scaling law (*Borghuis et al., 2008*). The fact that the two theories corroborate each other well is reassuring and suggests that the results are robust to the details of the analysis.

Our approach could be applied to analyzing the retinal mosaic characteristics of different animals. Adult zebrafish, for example, feature a highly regular mosaic with fixed 2:2:1:1 R:G:B:U cone ratios (*Engström, 1960*). Since our analysis has highlighted the importance of prior statistics in determining the optimal design, one might speculate whether this regularity results from the particular visual world of zebrafish (i.e. underwater, low signal-to-noise ratio), which perhaps demands a more balanced ratio

of different cone types to achieve the maximum amount of information transmission. Further study that characterizes in detail the natural scene statistics of the zebrafish's environment might help us to better understand this question (*Zimmermann et al., 2018*; *Cai et al., 2020*). It would also be interesting to incorporate into the formulation an explicit specification of how the goal of vision might vary across species. One extension to the current approach to incorporate this would be to specify an explicit loss function for each species and find the reconstruction that minimizes the expected (over the posterior of images) loss (*Berger, 1985*), although implementing this approach would be computationally challenging. Related is the task-specific accuracy maximization analysis formulation (*Burge and Geisler, 2011*; see *Burge, 2020* for a review).

Second, we applied our framework to cone excitations of retinal mosaics with varying degrees of optical quality, photoreceptor size, density, and cone spectral sensitivity. The reconstructed images reflect accurately the information loss in the initial encoding, including spatial blur due to optical aberration and mosaic sampling, pixel noise due to Poisson variability in the cone excitations, and reduction of chromatic contrast in anomalous trichromacy. Although we have mainly focused on visualization of these effects in our current paper, it would be possible to perform quantitative analyses. In fact, our reconstruction algorithm could provide a natural 'front-end' extension to many image-based perceptual quality metrics, such as spatial CIELAB (*Zhang and Wandell, 1997*; *Lian, 2020*), structural similarity (*Wang et al., 2004*), low-level feature similarity (FSIM; *Zhang et al., 2011*), or neural network-based approaches (*Bosse et al., 2018*). Doing so would incorporate factors related to the initial visual encoding explicitly into the resulting image quality metrics.

In addition, when SNR is high, we found that we are able to fully recover color information even from an anomalous trichromatic mosaic. As SNR drops, this becomes less feasible. Although our analysis does underestimate the amount of total noise in the visual system (i.e. we only consider noise at cone excitations, but see *Ala-Laurila et al., 2011* for a detailed treatment of noise in the retina), this nonetheless suggests that a downstream circuit that properly compensates for the shift in cone spectral sensitivity can, in principle, maintain relatively normal color perception in the low noise regime (*Tregillus et al., 2021*). This may potentially be related to some reports of less than expected difference in color perception between anomalous trichromats and color normal observers (*Bosten, 2019*; *Lindsey et al., 2020*).

Third, we speculate that image reconstruction could provide a reasonable proxy for modeling percepts in various psychophysical experiments. We found that images reconstructed from dichromatic mosaics resemble results generated by previously proposed methods for visualizing dichromacy, including one that uses explicit knowledge of dichromatic subjects' color appearance reports (*Brettel et al., 1997*). We have also reproduced the 'zebra stripes' and 'orientation reversal' aliasing patterns when reconstructing images from cone excitations to spatial frequencies above the mosaic Nyquist limit, similar to what has been documented experimentally in human subjects (*Williams, 1985*; *Coletta and Williams, 1987*). In a similar vein, previous work has used a simpler image reconstruction method to model the color appearance of small spots light stimulus presented to single cones using adaptive optics (*Brainard et al., 2008*). Our method could also be applied to such questions, and also to a wider range of adaptive optics (AO) experiments (e.g. *Schmidt et al., 2019*; *Neitz et al., 2020*), to help understand the extent to which image reconstruction can capture perceptual behavior. More speculatively, it may be possible to use calculations performed within the image reconstruction framework to synthesize stimuli that will maximally discriminate between different hypothesis about how the excitations of sets of cones are combined to form percepts, particularly with the emergence of technology that enables precise experimental control over the stimulation of individual cones in human subjects (*Harmening et al., 2014*; *Sabesan et al., 2016*; *Schmidt et al., 2019*).

Last, we showed that our method can be used in conjunction with analysis of psychophysical discrimination performance, bringing to this analysis the role of statistical regularities of natural images. In our initial exploration, we found that the image-reconstruction based observer exhibits significant interaction between spatial frequency and chromatic direction in its contrast sensitivity function, a behavior distinct from its Poisson ideal observer counterpart, and is more similar to the human observer. Future computations will be needed to understand in more detail whether the reconstruction approach can account for other features of human psychophysical discrimination performance that are not readily explained by ideal-observer calculations applied to the cone excitations.

Our current model only considers the representation up to and including the excitations of the cone mosaic. Post-excitation factors (e.g. retinal ganglion cells), especially in the peripheral visual field, are likely to lead to additional information loss. In this regard, we are eager to incorporate realistic models of retinal ganglion cells into the ISETBio pipeline. Nevertheless, the value of the analysis we have presented is to elucidate exactly what phenomena can or cannot be attributed to factors up to the cone excitations, thus helping to dissect the role of different stages of processing in determining behavior. For example, we found there is desaturation of chromatic content in reconstructed images in the periphery, with the details depending on interactions between the physiological optics, cone mosaic sampling, macular pigment density, and the model of natural image statistics. This is in contrast to more traditional explanations of the decrease in peripheral chromatic sensitivity, which often consider it in the context of models of how different cone types are wired to retinal ganglion cells (e.g. *Lennie et al., 1991*; *Mullen and Kingdom, 1996*; *Hansen et al., 2009*; *Field et al., 2010*; *Wool et al., 2018*). Whether the early vision factors are sufficient to account for the full variation in chromatic sensitivity awaits a more detailed future study, but the fact that early vision factors can play a role through their effect on the available chromatic information is a novel insight that should be incorporated into thinking about the role of post-excitation mechanisms.

More generally, we can consider the locus of the signals analyzed in the context of the encoding-decoding dichotomy of sensory perception (*Stocker and Simoncelli, 2006*; *Rust and Stocker, 2010*). Here, we reconstruct images from cone excitations, thus post-excitation processing may be viewed as part of the brain's implementation of the reconstruction algorithm. When we apply such an algorithm to, for example, the output of retinal ganglion cells, we shift the division. Our view is that analyses at multiple stages are of interest, and eventual comparisons between them are likely to shed light on the role of each stage.

Our current model also does not take into account fixational eye movements, which displace the retinal image at a time scale shorter than the integration period we have used here (*Martinez-Conde et al., 2004*; *Burak et al., 2010*). It has been shown that these small eye movements can increase psychophysically-measured visual acuity relative to that obtained with retinally-stabilized stimuli (*Rucci et al., 2007*; *Ratnam et al., 2017*). An intuition behind this is that fixational eye movements can increase the effective cone sampling density, if the visual system can sensibly combine information obtained across multiple fixation locations. This intuition is supported by computational analyses that integrate information across fixations while simultaneously estimating the eye movement path (*Burak et al., 2010*; *Anderson et al., 2020*). In their analysis, *Burak et al., 2010* showed the effectiveness of their algorithm depended both on the integration time of the sensory units whose excitations were processed, and also on the receptive field properties of those units. In addition, consideration of the effects of fixational eye movement might also benefit from an accurate model of the temporal integration that occurs within each cone, as a consequence of the temporal dynamics of the phototransduction cascade (*Angueyra and Rieke, 2013*). ISETBio in its current form implements a model of the phototransduction cascade as well as of fixational eye movements (see *Cottaris et al., 2020*). Future work should be able to extend our current results through the study of dynamic reconstruction algorithms within ISETBio.

Since our framework is centered on image reconstruction, one may naturally wonder whether we should have applied the more 'modern' technique of convolutional neural networks (CNNs), which have become the standard for image processing-related tasks (*Krizhevsky et al., 2012*). For our scientific purposes, the Bayesian framework offers an important advantage in its *modularity*, namely, the likelihood and prior are two separate components that can be built independently. This allows us to easily isolate and manipulate one of them (e.g. likelihood) while holding the other constant (e.g. prior), something we have done throughout this paper. In addition, building the likelihood function (i.e. render matrix $R$, see Materials and methods) is a forward process that is computationally very efficient. Performing a similar analysis with the neural network approach (or supervised learning in general) would require re-training of the network with a newly generated dataset (i.e. cone excitations paired with the corresponding images) for *every* condition in our analysis.

However, the ability of neural networks to represent more complex natural image priors (*Ulyanov et al., 2018*; *Kadkhodaie and Simoncelli, 2021*) is of great interest. Currently, we have chosen a rather simple, parametric description of natural image statistics, which leads to a numerical MAP solution. Previous work has proposed methods that alternate, within each iteration, between regularized

reconstruction and denoising, which effectively allow for transfer of the prior implicit in an image denoiser (e.g. a deep neural network denoiser) to be applied to any other domain with a known likelihood model (*Venkatakrishnan et al., 2013*; *Romano et al., 2017*). More recently, *Kadkhodaie and Simoncelli, 2021* developed a related but more explicit and direct technique to extract the image prior (a close approximation to the gradient of the log-prior density, to be precise) from a denoising deep neural network, which could be applied to our image reconstruction problem. We think this represents a promising direction, and in the future plan to incorporate more sophisticated priors, to evaluate the robustness of our conclusions to variations and improvements in the image prior.

To conclude, we believe our method is widely applicable to many experiments (e.g. adaptive optics psychophysics) designed for studying the initial visual encoding, for modeling the effect of changes of various components in the encoding process (e.g. in clinical conditions), and for practical applications (e.g. perceptual quality metric) in which the initial visual encoding plays an important role.

## Materials and methods

The problem of reconstructing images from neural signals can be considered in the general framework of estimating a signal $x$, given an (often lower-dimensional and noisy) measurement $m$. We take a Bayesian approach. Specifically, we model the generative process of measurement as the conditional probability $p(m|x)$ and the prior distribution of the signal as the probability density $p(x)$. We then take the estimate of the signal, $\hat{x}$, as the maximum a posteriori estimate $argmax\ p(m|\hat{x})\,p(\hat{x})$. We next explain in detail how each part of the Bayesian estimate is constructed.

### Likelihood function

In our particular problem, $x$ is a column vector containing the (vectorized) RGB pixel values of an input image of dimension $N*N*3$, where $N$ is the linear pixel size of the display. Below we will generalize from RGB images to hyperspectral images. The column vector $m$ contains the excitations of the $M$ cone photoreceptors. The relationship between $x$ and $m$ is modeled by the ISETBio software (*Cottaris et al., 2019*; *Cottaris et al., 2020*; *Figure 1*). ISETBio simulates in detail the process of displaying an image on the monitor, the wavelength-dependent optical blur of the human eye and spectral transmission through the lens and the macular pigment, as well as the interleaved sampling of the retinal image by the L, M and S cone mosaic. For the majority of simulations presented in our paper, we simulate a 1 deg foveal retina mosaic, which contains approximately 11,000 cone photoreceptors. A stochastic procedure was used to generate approximately hexagonal mosaics with eccentricity-varying cone density matched to that of the human retina (*Curcio et al., 1990*). See *Cottaris et al., 2019* for a detailed description of the algorithm. We use a wavelength-dependent point spread function empirically measured in human subjects (*Marimont and Wandell, 1994*; *Cottaris et al., 2019*), with a pupil size of 3 mm. We took the cone integration time to be 50 ms. The input images of size $128*128*3$ were displayed on a simulated typical CRT monitor (simulated with a 12 bit-depth in each of the RGB channels to avoid quantization artifacts).

Once the RGB pixel values in the original image are linearized, all the processes involved in the relation between $x$ and $m$, including image formation by the optics of the eye and the relation between retinal irradiance and cone excitations, are well described as linear operations. Furthermore, the instance-to-instance variability in cone excitations is described by a Poisson process acting independently in each cone. Thus $p(m|x)$ is the product of Poisson probability mass functions, one for each cone, with the Poisson mean parameter $\lambda_i$ for each cone determined by a linear transformation of the input image $x$. We describe the linear transformation between $x$ and the vector of Poisson mean parameters $\lambda$ by a matrix $R$, and thus obtain:

$$p(m|x) = \prod_{i=1}^{M} Poisson\left(m_i \mid \lambda_i = [Rx]_i\right).$$

We refer to the matrix $R$ as the *render matrix*. This matrix together with the Poisson variability encapsulates the properties of the initial visual encoding through to the level of the cone excitations. In cases where we parameterize properties of the initial visual encoding (parameters denoted by $\theta$ in the main text above), the render matrix is a function of these parameters.

Although ISETBio can compute the relation between the linearized RGB image values at each pixel and the mean excitation of each cone, it does so in a general way that does not exploit the linearity

of the relation. To speed the computations, we use ISETBio to precompute $R$. Each column of $R$ is a vector of mean cone excitations $r_j$ to a basis image $x_j$ with one entry set to one and the remaining entry set to zero. To determine $R$, we use ISETBio to compute explicitly each of its columns $r_j$. We verified that calculating mean cone excitations from an image via $Rx$ yields the same result as applying the ISETBio pipeline directly to the image.

See Code and data availability for parameters used in the simulation including display specifications (i.e. RGB channel spectra, display gamma function) and cone mosaic setup (i.e. cone spectral sensitivities, lens pigment and macular pigment density and absorption spectra), as well as some of the pre-computed render matrices.

## Null space of render matrix

To understand the information lost between an original RGB image and the mean cone excitations, we can take advantage of the linearity property of the render matrix. Variations in the image space that are within the null space of the (low-rank) render matrix $R$ will have no effect on the likelihood. That is, the cone excitation pattern provides no information to disambiguate between image variants that differ only by vectors within the null space of $R$. To obtain the null space of $R$, we used MATLAB function *null*, which computes the singular value decomposition of $R$. The set of right singular vectors whose associated singular values are 0 form a basis for the null space.

As an illustration, we generated random samples of images from the null space by taking linear combinations of its orthonormal basis vectors, where the weights are sampled independently from a Gaussian distribution with a mean of 0 and a standard deviation of 0.3. As shown in *Figure 3D*, altering an image by adding to it samples from the null space has no effect on the likelihood.

## Prior distribution

We also need to specify a prior distribution $p(x)$. The problem of developing statistical models of natural images has been studied extensively using numerous approaches, and remains challenging (*Simoncelli, 2005*). The high-dimensionality and complex structure of natural images makes it difficult to determine a high-dimensional joint distribution that properly captures the various forms of correlation and higher-order dependencies of natural images. Here, we have implemented two relatively simple forms of $p(x)$.

We first introduce a simple Gaussian prior $p(x)$ to set up the basic concepts and notations for image prior based on basis vectors. In particular, for the Gaussian prior, we assume $p(x) = \mathcal{N}(x|\mu, \Sigma)$. For convenience, we zero-centered our images when building priors, making $\mu = 0$. The actual mean value of each pixel is added back to each image when computing the likelihood and at the end of the reconstruction procedure. The covariance matrix $\Sigma$ can be estimated empirically, from a large dataset of natural images. Note that we can write the covariance matrix as its eigen-decomposition: $\Sigma = Q\Lambda Q^{-1}$. Defining $\beta = \Lambda^{-1/2}Q^{-1}x$, we have:

$$p(\beta) = \mathcal{N}(\beta \,|\, 0, \, I).$$

This derivation provides a convenient way of expressing our image prior: We can project images onto an appropriate set of basis vectors, and impose a prior distribution on the projected coefficients. In the case above, if we choose the basis vectors as the column vectors of $\Lambda^{-1/2}Q^{-1}$, we obtain an image prior by assuming that the entries of $\beta$ are each independently distributed as a univariant standard Gaussian (*Simoncelli, 2005*). Such a Gaussian prior can describe the first and second order statistics of natural images, but fails to capture important higher order structure (*Portilla et al., 2003*).

Our second model of $p(x)$ emerges from the basis set formulation. Rather than choosing the basis vectors from the eigen-decomposition as above and using a Gaussian distribution over the weights $\beta$, we instead choose an over-complete set of basis vectors using independent components analysis, and model the distribution of the entries of weight vector $\beta$ using the long-tailed distribution Laplace distribution. This leads to a sparse coding model of natural images (*Olshausen and Field, 1996*; *Simoncelli and Olshausen, 2001*). More specifically, we learned a set of $K$ $\left(K \geq 3N^2\right)$ basis vectors that lead to a sparse representation of our image dataset, through the reconstruction independent

component analysis (RICA) algorithm (*Le et al., 2011*) applied to whitened images, and took these as the columns of the basis matrix $E$. Our image prior in this case can be written as $p\left(\beta\right)$, with $\beta = E^+x$. Here $E^+$ represents the pseudoinverse of matrix $E$, and

$$p\left(\beta\right) = \prod_{k=1}^{K} \frac{1}{2b} \, exp\left(-\left|\beta_k\right|/b\right).$$

Note that we further scaled each column of $E$ to equalize the variance across $\beta_k$'s.

Both methods outlined above can be applied directly to small image patches. They are computationally intractable for larger images, however, since the calculation of basis vectors will involve either an eigen-decomposition of a large covariance matrix or independent component analysis of a set of high-dimensional image vectors. To address this limitation, we iteratively apply the prior distributions we have constructed above to overlapping small patches of the same size within a large image (*Guleryuz, 2006*).

To illustrate the idea, consider the following example: Assume we have constructed a prior distribution $p\left(y\right)$, for small image patches $y$ of size $N_{patch} * N_{patch}$. To model a larger image $x$ of size $pN_{patch} * pN_{patch}$, we could consider viewing $x$ as composed of $p*p$ independent patches of non-overlapping $y$'s. Under this assumption, the prior on $x$ could be expressed as the product:

$$p\left(x\right) \propto \prod_{j=1}^{p*p} p\left(y_j\right),$$

where $y_j$'s describe individual patches of size $N_{patch} * N_{patch}$ within $x$. The independence assumption is problematic, however, since $y_j$'s are far from independently sampled natural images: they need to be combined into a single coherent large image. Using this approach to approximate a prior would create block artifacts at the patch boundaries.

The basic idea above, however, can be extended heuristically to solve the block artifact problem by allowing $y_j$'s to overlap with each other. The degree of overlap can be viewed as an additional parameter of the prior, which we refer to here as the stride. This effectively implements a convolutional form of the sparse coding prior (*Gu et al., 2015*). Again, for example, consider a large image $x$ of size $pN_{patch} * pN_{patch}$. A stride of 1 will tile through all $\left(pN_{patch} - N_{patch} + 1\right) * \left(pN_{patch} - N_{patch} + 1\right)$ possible patches of size $N_{patch} * N_{patch}$ within $x$, yielding a prior distribution of the form:

$$p\left(x\right) \propto \prod_{j=1}^{\left(pN_{patch}-N_{patch}+1\right)*\left(pN_{patch}-N_{patch}+1\right)} p\left(y_j\right).$$

Although this form of prior is still an approximation, we have found it to work well in practice, and using it does not lead to visible block artifacts as long as the stride parameter is sufficiently smaller than $N_{patch}$.

## Maximum a posteriori estimation

To reconstruct the image $\hat{x}$ given a pattern of cone excitation $m$, we find the maximum a posteriori estimate: $\hat{x} = argmax \; p\left(m|\hat{x}\right) p\left(\hat{x}\right)$. In practice, this optimization is usually expressed in terms of its logarithmic counterpart: $\hat{x} = argmax \; \left[\log p\left(m|\hat{x}\right) p\left(\hat{x}\right) + \log p\left(\hat{x}\right)\right]$.

For the Poisson likelihood and sparse coding prior, the equation above becomes:

$$\hat{x} = argmax \; \left[\sum_{i=1}^{M} \left(-\lambda_i + m_i * log\left(\lambda_i\right)\right) + \gamma * \sum_{j=1}^{J} \sum_{k=1}^{K} \left|\beta_{jk}\right| + c\right]$$

where $\lambda = R\hat{x}$, $\beta_j = E^+y_j$, $y_j$'s are individual patches of size $N_{patch} * N_{patch}$ within $\hat{x}$. Each $\beta_j$ is of length $K$ and there are a total of $J$ (overlapping) patches. Lastly, $c$ is a constant that does not depend on $\hat{x}$.

In principle, the value of $\gamma$ can be analytically derived based on the parametric form of the prior. However, due to the approximate nature of our prior, introduced especially by the aggregation over patches, we left $\gamma$ as a free parameter. Treating $\gamma$ as a free parameter also provides some level of robustness against misspecification of the prior more generally. For most of the reconstruction results presented in this paper, the value of $\gamma$ was determined by maximizing reconstruction performance with a cross-validation procedure (see *Figure 2*). We also found that the optimal $\gamma$ values were similar across the two loss functions we considered. Note that the additional flexibility provided by this $\gamma$ parameter also provides us with a parametric way to manipulate and isolate the relative contribution

of the log-likelihood and log-prior terms to the reconstruction (e.g. *Figure 2*; also compare *Figure 7* and *Figure 7—figure supplement 1*).

The optimization problem required to obtain $\hat{x}$ can be solved efficiently using the MATLAB function *fmincon* by providing the analytical gradient to the minimization function:

$$\frac{\partial\ log\ p\ (m|x)}{\partial x} = \left(-1 + m \circ \frac{1}{\lambda}\right)^T * R,$$

$$\frac{\partial\ log\ p\ (y)}{\partial y} = sign\left(\beta\right)^T * E^+.$$

where $\lambda = Rx$, $\beta = E^+y$, $\circ$ denotes element-wise product between two vectors, $\frac{1}{\lambda}$ is the element-wise inverse of vector $\lambda$, and

$$\text{sign}\left(\beta_i\right) = \begin{cases} -1, & \beta_i < 0 \\ 1, & \beta_i > 0 \end{cases}.$$

### RGB image dataset

We used the ImageNet ILSVRC (*Russakovsky et al., 2015*) as our dataset for natural RGB images. Fifty randomly sampled images were reserved as the evaluation set, and the rest of the images were used for learning the prior and for cross-validation. For the sparse prior, we constructed a basis set size of $K = 768$, on image patches of size $16 * 16$ sampled from the training set, and used a stride of 4 when tiling larger images. We randomly sampled 20 patches from each one of the 5000 images in the training set for learning the prior (ICA analysis), and 500 images for the cross-validation procedure to determine the $\gamma$ parameter.

In our work, we simulate display of the RGB images on idealized monitor to generate spectral radiance as a linear combination of the monitor's RGB channel spectra. Thus, a prior over the linear RGB pixels values induces a full spatial-spectral prior. To make sure the constraints introduced by RGB images together with the monitor do not influence our results, we also conducted a control analysis using hyperspectral images directly, as described in the following section.

### Hyperspectral images

As a control analysis, we developed priors and reconstructed images directly on small patches of hyperspectral images. The development is essentially the same as above, with the generalization being to increase the number of channels in the images from 3 to $N$. In addition, since our algorithm treats images as high-dimensional vectors, it can be directly applied to reconstruct hyperspectral images. Here, we used images from *Nascimento et al., 2002* and *Chakrabarti and Zickler, 2011*. The dataset of *Nascimento et al., 2002* was pre-processed following the instructions provided by the authors, and the images of *Chakrabarti and Zickler, 2011* were converted to spectral radiance using the hyperspectral camera calibration data provided in that work. We further resampled the combined image dataset with a patch size of $18 * 18$ and 15 uniformly spaced wavelengths between 420 nm and 700 nm for a dataset of ~5000 patches. We retained 300 of them as the evaluation set, and the rest for prior learning and cross-validation. The remaining of the analysis (i.e. prior and reconstruction algorithm) followed the same procedures as those used for the RGB images, using number of basis functions $K = 4860$ and applied directly to each small image without the patchwise procedure.

See Code and data availability for the curated RGB and hyperspectral image dataset, as well learned basis functions for each sparse prior.

### Gaussian prior for synthetic images

We also reconstructed multivariate Gaussian distributed synthetic images with known chromatic and spatial correlations that we can explicitly manipulate (*Figure 5*). To construct these signals $x \sim \mathcal{N}\left(\mu,\ \Sigma\right)$, where $x$ is RGB image of size $N * N * 3$ ($N = 36$ in our current analysis), we set $\mu = 0.5$, and used a separable $\Sigma$ along its two spatial dimensions and one chromatic dimension. That is:

$$\Sigma = \Sigma_c \otimes \Sigma_s \otimes \Sigma_s,$$

where $\Sigma_c$ is the chromatic covariance matrix of size $(3 * 3)$:

$$\Sigma_c (i, j) = \sigma_c^2 * \rho_c^{|i-j|},$$

and $\Sigma_s$ is the spatial covariance matrix of size $(N * N)$:

$$\Sigma_s (i, j) = \sigma_s^2 * \rho_s^{|i-j|}.$$

In the covariance matrix constructions, $i, j$ index into entries of $\Sigma_c$ and $\Sigma_s$ at $i$-th row and $j$-th column. Here $\otimes$ represents the Kronecker product, thus producing the signal covariance matrix $\Sigma$ of size $(3N^2 * 3N^2)$ (**Brainard et al., 2008**; **Manning and Brainard, 2009**).

The parameters $\sigma_c^2$ and $\sigma_s^2$ determine the overall variance of the signal, which are fixed across all simulations, whereas by changing the value of $\rho_c$ and $\rho_s$, we manipulate the degree of spatial and chromatic correlation presented in the synthetic images (**Figure 5**).

We introduce an additional simplification for the case of reconstructions with respect to the synthetic Gaussian prior: We approximated the Poisson likelihood function with a Gaussian distribution with fixed variance. Thus, the reconstruction problem can be written as:

$$p(\beta) = \mathcal{N}(\beta \,|\, 0, \; I),$$

$$p(m|\beta) = \mathcal{N}\left(m \,|\, RQ\Lambda^{1/2}\beta, \; \sigma^2 I\right)$$

where $R$ is the render matrix, and $\Sigma = Q\Lambda Q^{-1}$.

The reconstruction problem with Gaussian prior and Gaussian noise matches the ridge regression formulation, and can be solved analytically by the regularized normal equations, applied directly to each small image without the patchwise procedure. Denote the design matrix $D = RQ\Lambda^{1/2}$:

$$\hat{\beta} = \left(D^T D + \gamma I\right)^{-1} D^T m$$

$$\hat{x} = Q\Lambda^{1/2}\hat{\beta}$$

Note that the $\gamma$ parameter here is also determined through a cross-validation routine. We adopted this simplification (using Gaussian noise) for the simulation results in **Figure 5**, in order to make it computationally feasible to evaluate the average reconstruction error across a large number of synthetic image datasets.

## Variations in retinal cone mosaic

To simulate a dichromatic observer, we constructed retinal mosaics with only two classes of cones but with similar spatial configuration. To simulate the deuteranomalous observer, we shifted the M cone spectral sensitivity function, setting its peak at 550 nm instead of the typical 530 nm. In both cases, the likelihood function (i.e. render matrix $R$) was computed using the procedure described above and the same Bayesian algorithm was applied to obtain the reconstructed images.

In **Figure 6**, we also present the results of two comparison methods for visualizing dichromacy, those of **Brettel et al., 1997** and **Jiang et al., 2016**, both are implemented as part of ISETBio routine. To determine the corresponding dichromatic images, we first computed the LMS trichromatic stimulus coordinates of the linear RGB value of each pixel of the input image, based on the parameters of the simulated CRT display. LMS coordinates were computed with respect to the Stockman-Sharpe 2 deg cone fundamentals (**Stockman and Sharpe, 2000**). The ISETBio function *lms2lmsDichromat* was then used to transform these LMS coordinates according to the two methods (see a brief description in the main text). Lastly, the transformed LMS coordinates were converted back to linear RGB values, and gamma corrected before rendering.

To simulate retinal mosaics at different eccentricities, we constructed retinal mosaics with the appropriate photoreceptor size, density (**Curcio et al., 1990**), and physiological optics (**Polans et al., 2015**), and computed their corresponding render matrices. The same Bayesian algorithm was applied to obtain the reconstructed images.

To simulate the interferometric experimental conditions of **Williams, 1985**, we used diffraction-limited optics without longitudinal chromatic aberration (LCA) for the computation of the cone

excitations, but used the likelihood function with normal optics for the reconstruction. This models subjects whose perceptual systems are matched to their normal optics and assumes there is no substantial adaptation within the short time span of the experiment.

## Contrast sensitivity function

We compared the spatial Contrast Sensitivity Function (CSF) between a standard, Poisson 2AFC ideal observer, and an image reconstruction-based observer.

We simulated stimulus modulations in two chromatic contrast directions, L + M and L - M. Contrast was measured as the vector length in the L and M cone contrast plane at 5 spatial frequencies, $[2, 4, 8, 16, 32]$ cycles per degree. For each chromatic direction and spatial frequency combination, the sensitivity is defined as the inverse of threshold contrast.

We used the QUEST+ procedure (*Watson, 2017*) as implemented in MATLAB by Brainard (mQUESTPlus; https://github.com/BrainardLab/mQUESTPlus; *Brainard, 2022*) for estimating the simulated threshold efficiently as follows: We initialized the procedure with the contrast near the middle of a pre-defined possible stimulus range. For each contrast, we first generated a null template $T_{\text{null}}$, which is the noise-free, average excitations of a 0.5 deg foveal mosaic with $N_{\text{cones}}$ cones to a uniform background stimulus; and a target template $T_{\text{targ}}$, which is the noise-free, average cone excitations to a grating stimulus at that contrast level. We then simulated 128 two alternative forced choice (TAFC) trials at this contrast. For each trial, two Poisson-noise corrupted observed sets of cone excitations $r_{\text{null}}$ and $r_{\text{targ}}$, are generated based on $T_{\text{null}}$ and $T_{\text{targ}}$, respectively. We determine the accuracy of for TAFC trials with the target in the first interval. Based on the observer responses, the QUEST+ procedure chooses the next test contrast according to an information-maximization criterion (*Watson, 2017*). The process is repeated 15 times, for a total of 15 * 128 = 1920 trials.

For the Poisson TAFC observer, we directly compute the likelihood ratio for the two possible orderings of the null and target stimulus:

$$\Lambda = \frac{Poisson\left(r_{\text{targ}}|T_{\text{targ}}\right) \; Poisson\left(r_{\text{null}}|T_{\text{null}}\right)}{Poisson\left(r_{\text{targ}}|T_{\text{null}}\right) \; Poisson\left(r_{\text{null}}|T_{\text{targ}}\right)}.$$

Taking the logarithm of the equation above, the decision rule simplifies to the following:

$$d = \sum_{i=1}^{N_{\text{cones}}} \left\{ \left(r_{\text{targ}} \circ \log T_{\text{targ}} + r_{\text{null}} \circ \log T_{\text{null}}\right) - \left(r_{\text{null}} \circ \log T_{\text{targ}} + r_{\text{targ}} \circ \log T_{\text{null}}\right) \right\}_i$$

where $\circ$ denotes element-wise product between two vectors. The simulated observer correctly chooses target in first interval when $d > 0$, and incorrectly test in second when $d < 0$. Because of symmetry, we only need to simulated one of the two TAFC orders.

For the image reconstruction-based observer, given the cone responses, it first applies the reconstruction algorithm to obtain the image template $\hat{T}_{\text{null}}$ and $\hat{T}_{\text{targ}}$ from $T_{\text{null}}$ and $T_{\text{targ}}$, and also noisy image instances $\hat{r}_{\text{null}}$ and $\hat{r}_{\text{targ}}$ by applying the same algorithm to $r_{\text{null}}$ and $r_{\text{targ}}$. We then perform a template-matching decision rule as follows:

$$d = \sqrt{\|\hat{r}_{\text{targ}} - \hat{T}_{\text{targ}}\|_2^2 + \|\hat{r}_{\text{null}} - \hat{T}_{\text{null}}\|_2^2} - \sqrt{\|\hat{r}_{\text{null}} - \hat{T}_{\text{targ}}\|_2^2 + \|\hat{r}_{\text{targ}} - \hat{T}_{\text{null}}\|_2^2}$$

where $\| \cdot \|_2$ represents the $L_2$ norm of a vector. The template observer correctly chooses target in first interval when $d < 0$, and incorrectly target in second interval when $d > 0$. We choose the template matching procedure for computational convenience. Note that because the variability in the reconstructed images is not independent across pixels, this procedure is not ideal.

## Code and data availability

The MATLAB code used for this paper is available at: https://github.com/isetbio/ISETImagePipeline, (copy archieved at swh:1:rev:72e7296dcaf8ebdcca35776d7a98026c8f041427, *Zhang, 2022*).

In addition, the curated RGB and hyperspectral image datasets, parameters used in the simulation including display and cone mosaic setup, as well as the intermediate results such as the learned sparse priors, likelihood functions (i.e. render matrices), are available through: https://tinyurl.com/26r92c8y.

## Additional information

### Competing interests
Ling-Qi Zhang, Nicolas P Cottaris, David H Brainard: Funding provided by Facebook Reality Labs.

### Funding

| Funder | Grant reference number | Author |
|--------|------------------------|--------|
| Facebook Reality Labs | | Ling-Qi Zhang<br>Nicolas P Cottaris<br>David Brainard |

The funders had no role in study design, data collection and interpretation, or the decision to submit the work for publication.

### Author contributions
Ling-Qi Zhang, Conceptualization, Data curation, Formal analysis, Investigation, Methodology, Software, Validation, Visualization, Writing – original draft, Writing – review and editing; Nicolas P Cottaris, Data curation, Methodology, Software, Validation, Visualization, Writing – review and editing; David H Brainard, Conceptualization, Funding acquisition, Methodology, Project administration, Resources, Supervision, Validation, Writing – review and editing

### Author ORCIDs
Ling-Qi Zhang ![ORCID] http://orcid.org/0000-0001-8468-7927
Nicolas P Cottaris ![ORCID] http://orcid.org/0000-0003-1829-6340
David H Brainard ![ORCID] http://orcid.org/0000-0001-9827-543X

### Decision letter and Author response
Decision letter https://doi.org/10.7554/eLife.71132.sa1
Author response https://doi.org/10.7554/eLife.71132.sa2

## Additional files

### Supplementary files
• Transparent reporting form

### Data availability
The MATLAB code used for this paper is available at: https://github.com/isetbio/ISETImagePipeline, (copy archieved at swh:1:rev:72e7296dcaf8ebdcca35776d7a98026c8f041427). In addition, the curated RGB and hyperspectral image datasets, parameters used in the simulation including display and cone mosaic setup, as well as the intermediate results such as the learned sparse priors, likelihood functions (i.e., render matrices), are available through: https://tinyurl.com/26r92c8y.

The following previously published dataset was used:

| Author(s) | Year | Dataset title | Dataset URL | Database and Identifier |
|-----------|------|---------------|-------------|--------------------------|
| Chakrabarti A, Zickler T | 2011 | Real-World Hyperspectral Images Database | http://vision.seas.harvard.edu/hyperspec/download.html | Harvard School of Engineering and Applied Sciences, hyperspectral-realworld |

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
