## [Editor Report]

This rigorous computational study simulates the sampling of the visual image by cone photoreceptors in the human eye, and explains how the image content can be reconstructed from those cone signals. The authors show that a number of properties of the human retina and of human perception are predicted from these simulations. Their modeling framework also serves to unify previous treatments and invites extension to subsequent stages of visual processing.

---

## [Decision Letter]

**Decision letter after peer review:**

Thank you for submitting your article "An Image Reconstruction Framework for Characterizing Early Vision" for consideration by *eLife*. Your article has been reviewed by 3 peer reviewers, including Markus Meister as Reviewing Editor and Reviewer #1, and the evaluation has been overseen by Tirin Moore as the Senior Editor.

The reviewers have discussed their comments, and the Reviewing Editor has drafted this consensus review to help you prepare a revised submission.

Essential revisions:

Main recommendations

1. Sensor model: The cone mosaic is assumed to be totally regular (a triangular array) with irregularity in cone type. What happens to reconstruction when the array is not a perfect triangular array?

2. Image statistics model: Please comment on the role of fixational eye movements during human vision. Under normal viewing conditions the retinal image doesn't hold still for the 50 ms integration time used here (line 357). In reality the image drifts so fast that any given cone looks at different image pixels every 5 ms (e.g. doi.org/10.1073/pnas.1006076107). Please discuss how this might affect the conclusions derived from an assumption of static images.

3. The natural scenes prior: This is a prominent component of the algorithm. Please elaborate how important the inclusion that prior term is for producing the results. For example:

3.1. Figures6, 7, and 8: Some supplemental comparison images with no-prior or weak-prior estimates would be helpful to visualize the effect that including the prior term has on the results presented here.

3.2. Figure 9: How important is the natural scenes prior for replicating the gratings psychophysics results? If you used just an MLE estimate, or reduced the weight on the prior term, would the results change dramatically?

4. Extrafoveal vision: At the eccentricities considered in Figure 8 the circuitry of the retina already pools over many cones. Why is a reconstruction based on differentiable cones still relevant here? Generally some more discussion of post-receptor vision would be helpful, or at least a justification for not considering it.

5. Please offer some interpretation for the visualizations produced by the Bayesian model, for example:

5.1. In Figure 6, the protanopia images have a reddish hue, and the images generated using reference methods do not.

5.2. In Figure 7, the images tend to get more speckled as light intensity decreases, which doesn't seem to match up with perception during natural vision.

5.3. In Figure 8, we might expect from human vision that chromatic saturation would increase as we move to the periphery, but the example images don't show that.

6. Relation to prior work:

6.1. Discuss how the current assumptions differ from Garrigan et al., (2010).

6.2. Discuss relation to the Plug and Play Bayesian image reconstruction and image restoration methods (e.g. doi: 10.1109/TCI.2016.2629286, doi: 10.1109/TPAMI.2021.3088914). These methods are also optimization-based MAP estimation algorithms, and are conceptually quite similar to the approach taken in the paper.

6.3. Repeatedly the results of this new approach end up consistent with earlier work that operated with simpler analysis (lines 318, 435, 522). In the discussion, please give a crisp summary of what new insights came from the more complex approach.

6.4. When introducing ideas that are part of conventional wisdom, a broader list of citations would help the reader, for example: the notion that multi-chromatic receptors are less useful in dim light (line 347); the optimal allocation of spectral types given the spectra of natural scenes (line 235 ff); the importance of prior distributions in evaluating visual system design (line 277).

Other suggestions

7. Title and elsewhere: “Early vision” is often interpreted as “everything up to V1” (see textbooks and e.g. doi.org/10.1523/JNEUROSCI.3726-05.2005). Here the signal hasn’t even emerged from the receptors. None of the post-receptoral circuitry is included, which ultimately comes to dominate visual perception. Please consider a title that is more specific to the article.

8. Figure 5:

8.1. Maybe plot all curves on the same y-scale. Could be easier to see the systematic variation.

8.2. Maybe color the symbol nearest the minimum of each function.

9. Figure7:

9.1. Please include the original images, since in those panels the reader is trying to compare image degradation (also in S5).

9.2. What if at twilight the goal is to reconstruct the gray scale image and not the RBG image? Would the reconstruction be more spatially accurate and less noisy?

10. Lines 525-533. Other species like zebrafish have a much more limited range of tasks to perform than humans. Is image reconstruction still the appropriate cost function in those cases?

---

## [Author Response]

Essential revisions:Main recommendations1. Sensor model: The cone mosaic is assumed to be totally regular (a triangular array) with irregularity in cone type. What happens to reconstruction when the array is not a perfect triangular array?

We appreciate this comment, which revealed a lacuna in our exposition. Our calculations were actually performed with more realistic mosaics, but we did not describe this. We have now expanded the Methods to clarify, and to provide a reference to the procedure we used for mosaic generation.

Page 29, Line 792. “A stochastic procedure was used to generate approximately hexagonal mosaics with eccentricity-varying cone density matched to that of the human retina (Curcio et al., 1990). See Cottaris et al., (2019) for a detailed description of the algorithm.”

2. Image statistics model: Please comment on the role of fixational eye movements during human vision. Under normal viewing conditions the retinal image doesn't hold still for the 50 ms integration time used here (line 357). In reality the image drifts so fast that any given cone looks at different image pixels every 5 ms (e.g. doi.org/10.1073/pnas.1006076107). Please discuss how this might affect the conclusions derived from an assumption of static images.

How fixational eye movements interact with reconstruction is an interesting and topical question, and a full exploration within the ISETBio framework would represent a paperlength project in its own right. We agree that some discussion is warranted here, however, and have added the following to the discussion:

“Our current model also does not take into account fixational eye movements, which displace the retinal image at a time scale shorter than the integration period we have used here (Martinez-Conde, Macknik, and Hubel 2004; Burak et al., 2010). […] Future work should be able to extend our current results through the study of dynamic reconstruction algorithms within ISETBio.”

3. The natural scenes prior: This is a prominent component of the algorithm. Please elaborate how important the inclusion that prior term is for producing the results. For example:3.1. Figures6, 7, and 8: Some supplemental comparison images with no-prior or weak-prior estimates would be helpful to visualize the effect that including the prior term has on the results presented here.

We agree that further elaboration on the role of the prior is a good idea. We have added a short summary of the importance of the natural image prior early on, and now follow that in the paper with additional analyses and comment.

Page 5, Line 138. “In the context of our current study, the role of the natural image prior comes in several forms, as we will demonstrate in Results. First, since the reconstruction problem is underdetermined, the prior is a regularizer, providing a unique MAP estimate; Second, the prior acts as a denoiser, counteracting the Poisson noise in the cone excitation; Lastly, the prior guides the spatial and spectral demosaicing of the signals provided via the discrete sampling of the retinal image by the cone mosaic.”

As we previously demonstrated in Figures 2 and 3, due to the presence of cone noise, estimation without a prior (maximum likelihood estimation) is highly subject to the effects of noise fluctuations – the reconstruction tracks the noise. This observation applies to all of our analyses but is particularly pertinent to the results shown in Figure 7, since those center around the effects of varying the signal-to-noise ratio of the cone excitations. We have elaborated on this result in response to Comment 5.2 below with the addition of Figure 7-S1, please refer to that response for more detail.

In addition, we conducted additional analyses associated with Figure 8, where we explored the effect of the prior on reconstruction in the absence of cone noise, by providing MLE reconstructions for comparison with those in Figure 8. We think this analysis provides valuable additional insight. The newly added figure is Figure 8-S4. We added the following to the main text:

Page 18, Line 452. “Lastly, to emphasize the importance of the natural image prior, we performed a set of maximum likelihood reconstructions with no explicit prior constraint, which resulted in images with less coherent spatial structure and lower fidelity color appearance (Figure 8-S4). Thus, the prior here is critical for the proper demosaicing and interpolation of the information provided by the sparse cone sampling at these peripheral locations.”

However, an important caveat here is that for the reconstruction problem we consider, the MLE estimate is not unique: variations within the null space of the render matrix do not influence the likelihood (Figure 3). This ambiguity is particularly pertinent to the dichromatic reconstructions, as large differences in color appearance can occur within the null space of a dichromatic mosaic’s render matrix. In fact, we have verified numerically that the original color image, an MLE estimate from the dichromatic mosaic (without cone noise), and linear mixtures of the two under the constraint that the mixture weights sum to one, all have the same (maximum) likelihood value (see Author response image 1). Thus, having a prior is crucial for obtaining well-defined dichromatic reconstructions. Rather than adding the Reviewer Figure to the paper, however, we have added the following prose to the discussion of Figure 6: Page 14, Line 329. “Note that in the case where there is no simulated cone noise (as in Figure 6), the original image has a likelihood at least as high as the reconstruction obtained via our method. Thus, the difference between the original images and each of the corresponding dichromatic reconstructions is driven by the image prior. On the other hand, the difference in the reconstructions across the three types of dichromacy illustrates how the different dichromatic likelihood functions interact with the prior.”

**Author response image 1. sa2fig1:** A set of six images with the same (maximum) likelihood for a deuteranopic cone mosaic. The top-left image is the original image, the bottom-right image is one MLE estimate for the dichromatic mosaic (without cone noise), and the other four images are produced as linear mixtures of the two, with the mixture weights summing to one. Without an explicit prior constraint, all these 6 images (and many others with pixel differences in the null space of the render matrix) provide a valid MLE solution to the reconstruction problem.

Also see our response to Comment 5.1 for a discussion of the three methods we compared for visualizing dichromacy.

3.2. Figure 9: How important is the natural scenes prior for replicating the gratings psychophysics results? If you used just an MLE estimate, or reduced the weight on the prior term, would the results change dramatically?

This is also an excellent question. We have now added a supplementary figure (Figure 10-S1) showing that an observer that makes decisions based on a maximum-likelihood reconstruction using the same type of template-based decision rule as we used for the reconstruction-based CSF in Figure 10 will produce contrast sensitivity similar to the Poisson ideal observer, albeit at a lower overall sensitivity level (since the template based decision rule does not handle noise as well as the ideal observer). That is, the prior matters quite a bit here.

Page 22, Line 545. “We attribute these effects to the role of the image prior in the reconstructions, which leads to selective enhancement/attenuation of different image components. In support of this idea, we also found that an observer based on maximum likelihood reconstruction without the explicit prior term produced CSFs similar in shape to the Poisson ideal observer (Figure 10-S1).”

4. Extrafoveal vision: At the eccentricities considered in Figure 8 the circuitry of the retina already pools over many cones. Why is a reconstruction based on differentiable cones still relevant here? Generally some more discussion of post-receptor vision would be helpful, or at least a justification for not considering it.

We agree with the reviewer that post-receptoral factors are important to consider, both for foveal and for peripheral vision. In this regard, we are eager to expand our current model to include retinal ganglion cells. Nevertheless, we believe that there is considerable value of the analysis as we have developed and presented it here. Indeed, the current analysis elucidates what phenomenon can or cannot be attributed to factors up to and including the cone mosaic, and thus clarifies what phenomena require explanation by later stages of processing. We have added to the discussion on this point:

Page 26, Line 687. “Our current model only considers the representation up to and including the excitations of the cone mosaic. […] When we apply such an algorithm to, for example, the output of retinal ganglion cells, we shift the division. Our view is that analyses at multiple stages are of interest, and eventual comparisons between them are likely to shed light on the role of each stage.”

5. Please offer some interpretation for the visualizations produced by the Bayesian model, for example:5.1. In Figure 6, the protanopia images have a reddish hue, and the images generated using reference methods do not.

Thanks for the question. We have now extended the text to discuss the similarities and differences among the three methods in terms of how the color in the visualization is determined as follows (also see the last section of our response to Comment 3.1):

Page 14, Line 340. “To determine an image based on the excitations of only two classes of cones, any method will need to rely on a set of regularizing assumptions to resolve the ambiguity introduced by the dichromatic retinas. Brettel et al., (1997) started with the trichromatic cone excitations of each image pixel, and projected these onto a biplanar surface, with each plane defined by the neutral color axis and an anchoring stimulus identified through color appearance judgments made across the two eyes of unilateral dichromats. […] We find the general agreement between the reconstruction-based methods and the one based subject reports an encouraging sign that the reconstruction approach can be used to predict aspects of appearance.”

In addition, we also found that in our previous analysis, the assumed display when rendering the two comparison methods was a generic sRGB display, not the CRT display we have used in the reconstruction routine. We have fixed this and updated Figure 6, although this results in no noticeable difference in the visualization as far as we can tell. We expanded the Methods section to include the details of the implementation:

Page 35, Line 1050. “In Figure 6 we also present the results of two comparison methods for visualizing dichromacy, those of Brettel et al., (1997) and Jiang et al., (2016), both are implemented as part of ISETBio routine. To determine the corresponding dichromatic images, we first computed the LMS trichromatic stimulus coordinates of the linear RGB value of each pixel of the input image, based on the parameters of the simulated CRT display. LMS coordinates were computed with respect to the Stockman-Sharpe 2-deg cone fundamentals (Stockman and Sharpe 2000). The ISETBio function *lms2lmsDichromat* was then used to transform these LMS coordinates according to the two methods (see a brief description in the main text). Lastly, the transformed LMS coordinates were converted back to linear RGB values, and γ corrected before rendering.”

5.2. In Figure 7, the images tend to get more speckled as light intensity decreases, which doesn't seem to match up with perception during natural vision.

Thanks for this insightful observation, which led us to recheck our calculations. In the original simulations, there was an error where the value of prior weight λ we used was too small, thus leading to an overly weak prior. We redid these calculations with the weight correctly chosen via our cross-validation procedure and updated Figure 7. In the corrected version, the increase in noise reduces the amount of spatial detail in the reconstructed images due to the denoising effect of the image prior, but the images do not get more “speckled”. This is more consistent with intuition. The text has been updated as following:

Page 16, Line 387. “At lower intensities, however, the deuteranomalous reconstruction lacks chromatic content still present in the normal reconstruction (second and third row). The increase in noise also reduces the amount of spatial detail in the reconstructed images, due to the denoising effect driven by the image prior. Furthermore, a loss of chromatic content is also seen for the reconstruction from the normal mosaic at the lowest light level (last row).”

Further, we have included as a supplementary figure the simulations done with the original lower λ value, as the comparison demonstrates the effect of cone noise when the prior is underweighted, which is a useful point to make in response to Comment 3.1 above:

Page 17, Line 407. “The prior weight parameter in these set of simulations was set based on a cross-validation procedure that minimizes RMSE (λ = 0.05). To highlight interaction between noise and the prior, we have also included a set of reconstructions with the prior weight set to a much lower level (λ = 0.001), see Figure 7-S1.”

5.3. In Figure 8, we might expect from human vision that chromatic saturation would increase as we move to the periphery, but the example images don't show that.

Our reading is that previous literature tends to find a decrease in chromatic sensitivity at peripheral visual eccentricities, at least for the red-green axis of color perception and some stimulus spatial configurations. Thus, we think our simulation is consistent with the literature in that a desaturation of the reconstructed images is qualitatively akin to a decrease in chromatic sensitivity, albeit with the degree of desaturation depending on the details of the prior, optical blur, and cone mosaic. We have added the following additional text to the paper:

Page 18, Line 437. “In the image of the dragonfly, for example, the reconstructed colors are desaturated at intermediate eccentricities (e.g., Figure 8C, D), compared with the fovea (Figure 8A) and more eccentric locations (Figure 8E, F). The desaturation is qualitatively consistent with the literature that indicates a decrease in chromatic sensitivity at peripheral visual eccentricities, at least for the red-green axis of color perception and for some stimulus spatial configurations (Virsu and Rovamo 1979; Mullen and Kingdom 1996; but see Hansen, Pracejus, and Gegenfurtner 2009).”

On this general point, also see our response to Comment 4 above.

6. Relation to prior work:6.1. Discuss how the current assumptions differ from Garrigan et al., (2010).

Thanks for the suggestion. We have elaborated the Discussion section on differences between our method and the approach taken by Garrigan et al., (2010).

Page 24, Line 601. “Previous work (Garrigan et al., 2010) conducted a similar analysis with consideration of natural scene statistics, physiological optics, and cone spectral sensitivity, using an information maximization criterion. One advance enabled by our work is that we are able to fully simulate a 1-deg mosaic with naturalistic input, as opposed to the information-theoretical measures used by Garrigan et al., which became intractable as the size of the mosaic and the dimensionality of the input increased. In fact, Garrigan et al., (2010) approximated by estimating the exact mutual information for small mosaic size (N = 1 … 6 cones) and then extrapolated to larger cone mosaics using a scaling law (Borghuis et al., 2008). The fact that the two theories corroborate each other well is reassuring and suggests that the results are robust to the details of the analysis.”

6.2. Discuss relation to the Plug and Play Bayesian image reconstruction and image restoration methods (e.g. doi: 10.1109/TCI.2016.2629286, doi: 10.1109/TPAMI.2021.3088914). These methods are also optimization-based MAP estimation algorithms, and are conceptually quite similar to the approach taken in the paper.

Plug-and-Play and other related techniques (e.g., Alain and Bengio 2014; Romano, Elad, and Milanfar 2017), including one we cited previously (Kadkhodaie and Simoncelli, 2021), are related methods (see Introduction in Kadkhodaie and Simoncelli 2021 for a brief review) that enable transfer of the prior implicit in an image denoiser to other domains. We think these techniques represent a promising direction that should allow us to take advantage of the image priors learned by denoising convolution neural networks and apply them to our image reconstruction problem. We have expanded the Discussion section on these related techniques:

Page 28, Line 750. “However, the ability of neural networks to represent more complex natural image priors (Ulyanov, Vedaldi, and Lempitsky 2018; Kadkhodaie and Simoncelli 2021) is of great interest. […] We think this represents a promising direction, and in the future plan to incorporate more sophisticated priors, to evaluate the robustness of our conclusions to variations and improvements in the image prior.”

6.3. Repeatedly the results of this new approach end up consistent with earlier work that operated with simpler analysis (lines 318, 435, 522). In the discussion, please give a crisp summary of what new insights came from the more complex approach.

We agree that such a summary is useful. At a broad level, an important contribution of our work is that it unifies treatment of a diverse set of issues that have been studied in separate, although related ways. In this regard, the comparisons between our results and previous ones serves as an important validation of our approach. For novel results, we have included in the Discussion section a summary as follows:

Page 24, Line 573. “Our method enables both quantification and visualization of information loss due to various factors in the initial encoding, and unifies the treatment of a diverse set of issues that have been studied in separate, albeit related, ways. In several cases, we were able to extend previous studies by eliminating simplifying assumptions (e.g., by the use of realistic, large cone mosaics that operate on high-dimensional, naturalistic image input). To summarize succinctly, we highlight here the following novel results and substantial extensions of previous findings: (1) When considering the allocation of different cone types on the human retina, we demonstrated the importance of the spatial and spectral correlation structure of the image prior; (2) As we examined reconstructions as a way to visualize information loss, we observed rich interactions in how the appearances of the reconstruction vary with mosaic sampling, physiological optics, and the SNR of the cone excitations; (3) We found that the reconstructions are consistent with empirical reports of retinal spatial aliasing obtained with interferometric stimuli, adding an explicit image prior component and extending consideration of the interleaved nature of the trichromatic retinal cone mosaic relative to the previous treatment of these phenomena; (4) We linked image reconstructions to spatio-chromatic contrast sensitivity functions by applying a computational observer for psychophysical discrimination to the reconstructions. Below, we provide an extended discussion of key findings, as well as of some interesting open questions and future directions.”

The above noted, another important contribution of our work is that it allows for predictions of novel experiments. We have expanded on this point, just a little, in the discussion:

Page 26, Line 666. “Our method could also be applied to such questions, and also to a wider range of adaptive optics (AO) experiments (e.g., Schmidt et al., 2019; Neitz et al., 2020), to help understand the extent to which image reconstruction can capture perceptual behavior. More speculatively, it may be possible to use calculations performed within the image reconstruction framework to synthesize stimuli that will maximally discriminate between different hypothesis about how the excitations of sets of cones are combined to form percepts, particularly with the emergence of technology that enables precise experimental control over the stimulation of individual cones in human subjects (Harmening et al., 2014; Sabesan et al., 2016; Schmidt et al., 2019).”

6.4. When introducing ideas that are part of conventional wisdom, a broader list of citations would help the reader, for example: the notion that multi-chromatic receptors are less useful in dim light (line 347); the optimal allocation of spectral types given the spectra of natural scenes (line 235 ff); the importance of prior distributions in evaluating visual system design (line 277).

Thanks for the suggestions. We have included a broader list of citations at the three places mentioned above (Note that the line numbers have shifted from those in the comment, due to the revisions in the manuscript):

Page 9, Line 249. “Our results are in agreement with a previous analysis in showing that the empirically observed allocation of retinal photoreceptor type is consistent with the principle of optimal design (Garrigan et al., 2010; also see Levin et al., 2008; Manning and Brainard 2009; Tian et al., 2015; Jiang et al., 2017).”

Page 12, Line 294. “This analysis highlights the importance of considering visual system design in context of the statistical properties (prior distribution) of natural images, as it shows that the conclusions drawn can vary with these properties (Barlow 1961; Derrico and Buchsbaum 1991; Barlow and Földiàgk 1989; Atick, Li, and Redlich 1992; Lewis and Li 2006; Levin et al., 2008; Borghuis et al., 2008; Garrigan et al., 2010; Tkačik et al., 2010; Atick 2011; Burge 2020).”

Page 16, Line 392. “This observation may be connected to the fact that biological visual systems that operate at low light levels are typically monochromatic, potentially to increase the SNR of spatial vision at the cost of completely disregarding color (e.g., the monochromatic human rod system; see Manning and Brainard 2009 for a related and more detailed treatment; also see Walls 1942; Rushton 1962; Van Hateren 1993; Land and Osorio 2003).”

Other suggestions7. Title and elsewhere: “Early vision” is often interpreted as “everything up to V1” (see textbooks and e.g. doi.org/10.1523/JNEUROSCI.3726-05.2005). Here the signal hasn’t even emerged from the receptors. None of the post-receptoral circuitry is included, which ultimately comes to dominate visual perception. Please consider a title that is more specific to the article.

We agree with this comment, and have replaced all occurrences of “early vision” in the paper with either “the initial visual encoding” or “initial encoding”.

8. Figure 5:8.1. Maybe plot all curves on the same y-scale. Could be easier to see the systematic variation.8.2. Maybe color the symbol nearest the minimum of each function.

We have modified Figure 5 to explicitly mark the areas that are close to the minimum, which improves the presentation. We have also included a new supplementary figure for the same data but with matched y-axis. The main text is changed as follows:

Page 12, Line 286. “The dependence of the average reconstruction error on the L-cone proportion decreases as the chromatic correlation in the signal increases (Figure 5). A decrease of spatial correlation has little impact on the shape of the curves, but increases the overall magnitude of reconstruction error (Figure 5; to highlight the shape, the scale of the y-axis is different across rows and columns. See Figure 5-S1 for the same plot with matched y-axis scale). When both the chromatic and spatial correlation are high, there is a large margin of L-cone proportion within which the reconstruction error is close to the optimal (minimal) point (Figure 5, shaded area).”

9. Figure7:9.1. Please include the original images, since in those panels the reader is trying to compare image degradation (also in S5).

Thanks for the suggestions. We have now added the original images to these two figures to facilitate the comparison. Also note that S5 is now Figure 8-S3.

9.2. What if at twilight the goal is to reconstruct the gray scale image and not the RBG image? Would the reconstruction be more spatially accurate and less noisy?

We conducted an initial analysis to explore the possibility raised by this question. More specifically, we constrained the search space of the reconstruction algorithm to be grayscale images only (R = G = B at each pixel) and obtained the MAP estimate under this constraint. The prior weight was set to the same levels as Figure 7 (λ = 0.05) and Figure 7-S1 (λ = 0.001) in the main text. Visual examination did not reveal improvements in the quality of the reconstructed images (Author response image 2), with the most salient difference being the loss of the residual color in the images reconstructed under the grayscale constraint.

**Author response image 2. sa2fig2:** Grayscale image reconstruction from a normal trichromatic mosaic at twilight level, given two different prior weights. Compare to Figure 7 and Figure 7-S1 in the main text, we did not find meaningful improvements in the quality of the reconstructed images.

Finding the MAP estimate under the grayscale constraint is simple and numerically feasible. A more sophisticated method would involve first marginalizing the posterior. Concretely:

Define p(x) as a posterior over RGB images x given a pattern of cone excitations.

Define G as the transformation between any x and its corresponding grayscale image y (i.e. G could simply add the R, G, B values at each pixel location and divide by 3). Then, the posterior over is computed as:p(y)=∫p(y|x)p(x)dx=∫δ(y−Gx)p(x)dx,

where δ(⋅) is the vector-valued delta function. It is possible that the MAP estimate under this marginalized posterior would yield improved grayscale reconstructions.

Another quite interesting approach would be to provide an explicit loss function, and rather than choosing the MAP reconstruction, choose the reconstruction that minimizes the expected (over the posterior loss). The marginalization approach may be thought of as a special case of the loss function approach, where the loss function is set to be sensitive only to grayscale reconstruction error (e.g. L(x,x^)=||Gx−Gx^||2). We did introduce the idea of an explicit loss function in the paper (see Page 9), and now have added a note indicating the MAP estimate does not in general minimize the expected loss in Footnote 3 of Page 9.

The challenge of implementing the more sophisticated approaches described above, however, is that the integration over the high-dimensional p(x) is computationally intractable. Although various approximations exist in the literature, exploring those is beyond the scope of the current paper.

Manning and Brainard (2009, as cited in the manuscript) do treat in detail the closely related issue of how the optimal choice of photoreceptor mosaic varies with overall SNR, for a simplified model system that allowed exhaustive computational exploration using a reconstruction approach. Their conclusion is that the reason nocturnal visual systems typically utilize a single photoreceptor class is that one class of receptor will inevitably have better SNR than the others, and that as the overall SNR drops, the benefit of utilizing multiple receptor types to provide color vision is outweighed by the benefit of having all of the photoreceptors be of the class that has the best SNR. Conversely, at higher SNR a visual system can afford to intersperse additional receptor classes with lower SNR to gain the benefits of color vision. We think the reviewer may find that paper of interest, although it does not directly address the specific question raised here, of what would happen if the goal of vision changed as a function of how well that goal could be accomplished.

As interesting as we find this topic, in terms of the manuscript we have chosen to expand our discussion only slightly and point to a larger set of references (this passage also referenced in response to Comment 6.4 above), as we think going further will take the reader too far afield.

Page 16, Line 392. “This observation may be connected to the fact that biological visual systems that operate at low light levels are typically monochromatic, potentially to increase the SNR of spatial vision at the cost of completely disregarding color (e.g., the monochromatic human rod system; see Manning and Brainard 2009 for a related and more detailed treatment; also see Walls 1942; Rushton 1962; Van Hateren 1993; Land and Osorio 2003).”

10. Lines 525-533. Other species like zebrafish have a much more limited range of tasks to perform than humans. Is image reconstruction still the appropriate cost function in those cases?

We agree that cross-species differences in the tasks supported by visual perception are likely an important consideration. We think an interesting way to approach this in the long run would be to incorporate an explicit loss function into the formulation, and then consider what loss function might be most appropriate for each species under consideration (see discussion of loss functions in response to Comment 9.2 above). Beyond the computational challenges involved, doing this would also require detailed investigation about what the right loss function for a zebrafish is, and how that differs from the corresponding human loss function.

We have expanded the related Discussion section:

Page 25, Line 620. “Further study that characterizes in detail the natural scene statistics of the zebrafish’s environment might help us to better understand this question (Zimmermann et al., 2018; Cai et al., 2020). It would also be interesting to incorporate into the formulation an explicit specification of how the goal of vision might vary across species. One extension to the current approach to incorporate this would be to specify an explicit loss function for each species and find the reconstruction that minimizes the expected (over the posterior of images) loss (Berger 1985), although implementing this approach would be computationally challenging. Related is the task-specific accuracy maximization analysis formulation (Burge and Geisler 2011; see Burge 2020 for a review).”